# WHAT DO WE MEAN BY GENERALIZATION IN FEDERATED LEARNING?

**Honglin Yuan**[*]
Stanford University
`hongl.yuan@gmail.com`

**Warren Morningstar, Lin Ning, Karan Singhal**
Google Research
`{wmorning, linning, karansinghal}@google.com`

## ABSTRACT

Federated learning data is drawn from a distribution of distributions: clients are drawn from a meta-distribution, and their data are drawn from local data distributions. Thus generalization studies in federated learning should separate performance gaps from unseen client data (*out-of-sample gap*) from performance gaps from unseen client distributions (*participation gap*). In this work, we propose a framework for disentangling these performance gaps. Using this framework, we observe and explain differences in behavior across natural and synthetic federated datasets, indicating that dataset synthesis strategy can be important for realistic simulations of generalization in federated learning. We propose a semantic synthesis strategy that enables realistic simulation without naturally-partitioned data. Informed by our findings, we call out community suggestions for future federated learning works.

## 1 INTRODUCTION

Federated learning (FL) enables distributed clients to train a machine learning model collaboratively via focused communication with a coordinating server. In *cross-device* FL settings, clients are sampled from a population for participation in each round of training (Kairouz et al., 2019; Li et al., 2020a). Each participating client possesses its own data distribution, from which finite samples are drawn for federated training.

Given this problem framing, defining generalization in FL is not as obvious as in centralized learning. Existing works generally characterize the difference between empirical and expected risk for clients participating in training (Mohri et al., 2019; Yagli et al., 2020; Reddi et al., 2021; Karimireddy et al., 2020; Yuan et al., 2021). However, in cross-device settings, which we focus on in this work, clients are sampled from a large population with unreliable availability. Many or most clients may never participate in training (Kairouz et al., 2019; Singhal et al., 2021). Thus it is crucial to better understand expected performance for non-participating clients.

In this work, we model clients' data distributions as drawn from a meta population distribution (Wang et al., 2021), an assumption we argue is reasonable in real-world FL settings. We use this framing to define two generalization gaps to study in FL: the *out-of-sample gap*, or the difference between empirical and expected risk for participating clients, and the *participation gap*, or the difference in expected risk between participating and non-participating clients. Previous works generally ignore the participation gap or fail to disentangle it from the out-of-sample gap, but we observe significant participation gaps in practice across six federated datasets (see Figure 1), indicating that the participation gap is an important but neglected feature of generalization in FL.

We present a systematic study of generalization in FL across six tasks. We observe that focusing only on out-of-sample gaps misses important effects, including differences in generalization behavior across naturally-partitioned and synthetically-partitioned federated datasets. We use our results to inform a series of recommendations for future works studying generalization in FL.

---

[*]Work was completed while at Google Research.

Figure 1: **Federated training results demonstrating participation gaps for six different tasks.** We conduct experiments on four image classification tasks and two text prediction tasks. As described in Section 3.1, the participation gap can be estimated as the difference in metrics between *participating validation* and *unparticipating* data (defined in Figure 2). Prior works either ignore the participation gap or fail to separate it from other generalization gaps, indicating the participation gap is a neglected feature of generalization in FL.

**Our contributions:**

- Propose a *three-way split* for measuring out-of-sample and participation gaps in centralized and FL settings where data is drawn from a distribution of distributions (see Figure 2).
- Observe significant participation gaps across six different tasks (see Figure 1) and perform empirical studies on how various factors, *e.g.,* number of clients and client diversity, affect generalization performance (see Section 5).
- Observe significant differences in generalization behavior across naturally-partitioned and synthetically-partitioned federated datasets, and propose *semantic partitioning*, a dataset synthesis strategy that enables more realistic simulations of generalization behavior in FL without requiring naturally-partitioned data (see Section 4).
- Present a model to define the participation gap (Section 2), reveal its connection with data heterogeneity (Section 3.2), and explain differences in generalization behavior between label-based partitioning and semantic partitioning (Section 4.2).
- Present recommendations for future FL works, informed by our findings (see Section 6).
- Release an extensible open-source code library for studying generalization in FL (see Reproducibility Statement).

## 1.1 RELATED WORK

We briefly discuss primary related work here and provide a detailed review in Appendix A. We refer readers to Kairouz et al. (2019); Wang et al. (2021) for a more comprehensive introduction to federated learning in general.

**Distributional heterogeneity in FL.** Distributional heterogeneity is one of the most important patterns in federated learning (Kairouz et al., 2019; Wang et al., 2021). Existing literature on FL heterogeneity is mostly focused on the impact of heterogeneity on the training efficiency (convergence and communication) of federated optimizers (Karimireddy et al., 2020; Li et al., 2020b; Reddi et al., 2021). In this work, we identify that the participation gap is another major outcome of the heterogeneity in FL, and recommend using the participation gap as a natural measurement for dataset heterogeneity.

**Personalized FL.** In this work, we propose to evaluate and distinguish the generalization performance of clients participating and non-participating in training. Throughout this work, we focus on the classic FL setting (Kairouz et al., 2019; Wang et al., 2021) in which a single global model is learned from and served to all clients. In the *personalized* FL setting (Hanzely & Richtárik, 2020; Fallah et al., 2020; Singhal et al., 2021), the goal is to learn and serve different models for different clients. While related, our focus and contribution is orthogonal to personalization. In fact, our three-way split framework can be readily applied in various personalized FL settings. For example, for personalization via fine-tuning (Wang et al., 2019; Yu et al., 2020), the participating clients can be defined as the clients that contribute to the training of the base model. The participation gap can then be defined as the difference in post-fine-tuned performance between participating clients and unparticipating clients.

**Out-of-distribution generalization.** In this work, we propose to train models using a set of participating clients and examine their performance on heldout data from these clients as well as an additional set of non-participating clients. Because each client has a different data distribution, unparticipating clients' data exhibits distributional shift compared to the participating clients' validation data. Therefore, our work is related to the field of domain adaptation (Daumé III, 2009; Ben-David et al., 2007; Shimodaira, 2000; Patel et al., 2015), where a model is explicitly adapted to make predictions on a test set that is not identically distributed to the training set. The participation gap that we observe is consistent with findings from the out-of-distribution research community (Ovadia et al., 2019; Amodei et al., 2016; Lakshminarayanan et al., 2016), which shows on centrally trained (non-federated) models that even small deviations in the morphology of deployment examples can lead to systematic degradations in performance. Our setting differs from these other settings in that our problem framing assumes data is drawn from a distribution of client distributions, meaning that the training and deployment distributions eventually converge as more clients participate in training. In contrast, the typical OOD setup assumes that the distributions will never converge (since the deployment data is out-of-distribution, by definition it does not contribute to training). Our meta-distribution assumption makes the problem of generalizing to unseen distributions potentially more tractable.

## 2 SETUP FOR GENERALIZATION IN FL

We model each FL client as a data source associated with a local distribution and the overall population as a meta-distribution over all possible clients.

**Definition 2.1** (Federated Learning Problem). *1. Let $\Xi$ be the (possibly infinite) collection of all the possible data elements, e.g., image-label pairs. For any parameters $\boldsymbol{w}$ in parameter space $\Theta$, we use $f(\boldsymbol{w}, \xi)$ to denote the loss at element $\xi \in \Xi$ with parameter $\boldsymbol{w}$.*

*2. Let $\mathfrak{C}$ be the (possibly infinite) collection of all the possible clients. Every client $c \in \mathfrak{C}$ is associated with a local distribution $\mathcal{D}_c$ supported on $\Xi$.*

*3. Further, we assume there is a meta-distribution $\mathcal{P}$ supported on client set $\mathfrak{C}$, and each client $c$ is associated with a weight $\rho_c$ for aggregation.*

*The goal is to optimize the following two-level expected loss as follows:*

$$F(\boldsymbol{w}) := \mathbb{E}_{c \sim \mathcal{P}} \left[ \rho_c \cdot \mathbb{E}_{\xi \sim \mathcal{D}_c} [f(\boldsymbol{w}; \xi)] \right]. \tag{1}$$

Similar formulations as in Equation (1) have been proposed in existing literature (Wang et al., 2021; Reisizadeh et al., 2020; Charles & Konečný, 2020). To understand Equation (1), consider a random procedure that repeatedly draws clients $c$ from the meta-distribution $\mathcal{P}$ and then evaluates the loss on samples $\xi$ drawn from the local data distribution $\mathcal{D}_c$. Equation (1) then characterizes the weighted-average limit of the above process.

**Remark.** *The selection of client weights $\{\rho_c : c \in \mathfrak{C}\}$ depends on the desired aggregation pattern. For example, setting $\rho_c \equiv 1$ will equalize the performance share across all clients. Another common example is setting $\rho_c$ to be proportional to the training dataset size contributed by client $c$.*

**Intuitive Justification.** The formulation in Equation (1) is especially natural in cross-device FL settings, where the number of clients is generally large and modeling clients' local distributions as sampled from a meta-distribution is reasonable. This assumption also makes the problem of generalization to non-participating client distributions more tractable since samples from the meta-distribution are seen during training.

**Discretization.** While the ultimate goal is to optimize the expected loss over the entire meta-distribution $\mathcal{P}$ and client local distributions $\{\mathcal{D}_c : c \in \mathfrak{C}\}$, only finite training data and a finite number of clients are accessible during training. We call the subset of clients that contributes training data the **participating clients**, denoted as $\hat{\mathfrak{C}}$. We assume $\hat{\mathfrak{C}}$ is drawn from the meta-distribution $\mathcal{P}$. For each participating client $c \in \hat{\mathfrak{C}}$, we denote $\hat{\Xi}_c$ the training data contributed by client $c$. We call these data **participating training client data** and assume $\hat{\Xi}_c$ satisfies the local distribution $\mathcal{D}_c$.

**Definition 2.2.** *The empirical risk on the participating training client data is defined by*

$$F_{\mathrm{part\_train}}(\boldsymbol{w}) := \frac{1}{|\hat{\mathfrak{C}}|} \sum_{c \in \hat{\mathfrak{C}}} \left[ \rho_c \cdot \left( \frac{1}{|\hat{\Xi}_c|} \sum_{\xi \in |\hat{\Xi}_c|} f(\boldsymbol{w}; \xi) \right) \right]. \tag{2}$$

Equation (2) characterizes the performance of the model (at parameter $\boldsymbol{w}$) on the observed data possessed by observed clients.

There are two levels of generalization between Equation (2) and Equation (1): (i) the generalization from finite training data to unseen data, and (ii) the generalization from finite participating clients to unseen clients. To disentangle the effect of the two levels, a natural intermediate stage is to consider the performance on unseen data of participating (seen) clients.

**Definition 2.3.** *The semi-empirical risk on the participating validation client data is defined by*

$$F_{\mathrm{part\_val}}(\boldsymbol{w}) := \frac{1}{|\hat{\mathfrak{C}}|} \sum_{c \in \hat{\mathfrak{C}}} \left[ \rho_c \cdot (\mathbb{E}_{\xi \sim \mathcal{D}_c} f(\boldsymbol{w}; \xi)) \right]. \tag{3}$$

Equation (3) differs from Equation (2) by replacing the intra-client empirical loss with the expected loss over $\mathcal{D}_c$. We shall also call $F(\boldsymbol{w})$ defined in Equation (1) the **unparticipating expected risk** and denote it as $F_{\mathrm{unpart}}(\boldsymbol{w})$ for consistency. Now we are ready to define the two levels of generalization gaps formally.

**Definition 2.4.** *The out-of-sample gap is defined as $F_{\mathrm{part\_val}}(\boldsymbol{w}) - F_{\mathrm{part\_train}}(\boldsymbol{w})$.*

**Definition 2.5.** *The participation gap is defined as $F_{\mathrm{unpart}}(\boldsymbol{w}) - F_{\mathrm{part\_val}}(\boldsymbol{w})$.*

Note that these gaps are also meaningful in centralized learning settings where data is sampled from a distribution of distributions.

# 3 UNDERSTANDING GENERALIZATION GAPS

## 3.1 ESTIMATING RISKS AND GAPS VIA THE THREE-WAY SPLIT

Both $F_{\mathrm{part\_val}}$ and $F_{\mathrm{unpart}}$ take an expectation over the distribution of clients or data. To estimate these two risks in practice, we propose splitting datasets into three blocks. The procedure is demonstrated in Figure 2. Given a dataset with client assignment, we first hold out a percentage of clients (*e.g.,* 20%) as unparticipating clients, as shown in the rightmost two columns (in purple). The remaining clients are participating clients. We refer to this split as **inter-client split**. Within each participating client, we hold out a percentage of data (*e.g.,* 20%) as participating validation data, as shown in the upper left block (in orange). The remaining data is the participating training client data, as shown in the lower left block (in blue). We refer to this second split as **intra-client split**.

Figure 2: **Illustration of the three-way split via a visualization of the EMNIST digits dataset.** Each column corresponds to the dataset of one client. A dataset is split into participating training, participating validation, and unparticipating data, which enables separate measurement of out-of-sample and participation gaps (unlike other works). Note we only present the digit "6" for illustrative purposes.

Existing FL literature and benchmarks typically conduct either an inter-client or intra-client train-validation split. However, neither inter-client nor intra-client split alone can reveal the participation gap.[1] To the best of our knowledge, this is the first work that conducts *both* splits *simultaneously*.

## 3.2 WHY IS THE PARTICIPATION GAP INTERESTING?

**Participation gap is an intrinsic property of FL due to heterogeneity.** Heterogeneity across clients is one of the most important phenomena in FL. We identify that the participation gap is another

---

[1]To see this, observe that inter-client split can only estimate $F_{\mathrm{part\_train}}$ and $F_{\mathrm{unpart}}$, and intra-client split can only estimate $F_{\mathrm{part\_train}}$ and $F_{\mathrm{part\_val}}$.

outcome of heterogeneity in FL, in that the gap will not exist if data is homogeneous. Formally, we can establish the following proposition.

**Proposition 3.1.** *If $\mathcal{D}_c \equiv \mathcal{D}$ for any $c \in \mathfrak{C}$ and $\rho_c \equiv \rho$, then for any participating clients $\hat{\mathfrak{C}} \subset \mathfrak{C}$ and $\boldsymbol{w}$ in domain, the participation gap is always zero in that $F_{\mathrm{unpart}}(\boldsymbol{w}) \equiv F_{\mathrm{part\_val}}(\boldsymbol{w})$.*

Proposition 3.1 holds by definition as

$$F_{\mathrm{part\_val}}(\boldsymbol{w}) = \frac{1}{|\hat{\mathfrak{C}}|}\sum_{c\in\hat{\mathfrak{C}}}[\rho\cdot(\mathbb{E}_{\xi\sim\mathcal{D}_c}f(\boldsymbol{w};\xi))] = \rho\cdot(\mathbb{E}_{\xi\sim\mathcal{D}}f(\boldsymbol{w};\xi)) = \mathbb{E}_{c\sim\mathcal{P}}[\rho\cdot\mathbb{E}_{\xi\sim\mathcal{D}}[f(\boldsymbol{w};\xi)]] = F_{\mathrm{unpart}}(\boldsymbol{w}).$$

**Remark.** *We assume unweighted risk with $\rho_c \equiv \rho$ for ease of exposition. Even if $\rho_c$ are different, one can still show $\frac{F_{\mathrm{unpart}}(\boldsymbol{w})}{F_{\mathrm{part\_val}}(\boldsymbol{w})}$ is always equal to a constant independent of $\boldsymbol{w}$. Therefore the triviality of the participation gap for homogeneous data still holds in the logarithmitic sense.*

**Participation gap can quantify client diversity.** The participation gap can provide insight into a federated dataset since it provides a quantifiable measure of client diversity / heterogeneity. With other aspects controlled, a federated dataset with larger participation gap tends to have greater heterogeneity. For example, using the same model and hyperparameters, we observe in Section 5 that CIFAR-100 exhibits a larger participation gap than CIFAR-10. Unlike other indirect measures (such as the degradation of federated performance relative to centralized performance), the participation gap is intrinsic in federated datasets and more consistent with respect to training hyperparameters.

**Participation gap can measure overfitting on the population distribution.** Just as a generalization gap that increases over time in centralized training can indicate overfitting on training samples, a large or increasing participation gap can indicate a training process is overfitting on participating clients. We observe this effect in Figure 1 for Shakespeare and Stack Overflow tasks. Thus measuring this gap can be important for researchers developing models or algorithms to reduce overfitting.

**Participation gap can quantify model robustness to unseen clients.** From a modeler's perspective, the participation gap quantifies the loss of performance incurred by switching from seen clients to unseen clients. The smaller the participation gap is, the more robust the model might be when deployed. Therefore, estimating participation gap may guide modelers to design more robust models, regularizers, and training algorithms.

**Participation gap can quantify the incentive for clients to participate.** From a client's perspective, the participation gap offers a measure of the performance gain realized by switching from unparticipating (not contributing training data) to participating (contributing training data). This is a fair comparison since both $F_{\mathrm{part\_val}}$ and $F_{\mathrm{unpart}}$ are estimated on unseen data. When the participation gap is large (*e.g.,* if only a few clients participate), modelers might report the participation gap as a well-justified incentive to encourage more clients to join a federated learning process.

## 4    REFLECTIONS ON CLIENT HETEROGENEITY AND SYNTHETIC PARTITIONING

Since participation gaps can quantify client dataset heterogeneity, we study how participation gaps vary for different types of federated datasets. Many prior works (McMahan et al., 2017; Zhao et al., 2018; Hsu et al., 2019; Reddi et al., 2021) have created synthetic federated versions of centralized datasets. These centralized datasets do not have naturally-occurring client partitions and thus need to be synthetically partitioned into clients. Due to the importance of heterogeneity in FL, partitioning schemes generally ensure client datasets are heterogeneous in some respect. Previous works typically impose heterogeneity at the label level. For example, Hsu et al. (2019) create heterogeneous federated datasets by assigning each client a distribution over labels, where each local distribution is drawn from a Dirichlet meta-distribution. Once conditioned on labels, the drawing process is homogeneous. We refer to these schemes as **label-based partitioning**.[2]

While label heterogeneity is generally observed in natural federated datasets, it is not the *only* observed form of heterogeneity. In particular, each client in a natural federated dataset has its own

---

[2]To avoid confusion, throughout this work, we use the term "partition" to refer to assigning data with no client assignment into synthetic clients. The term "split" refers to splitting a federated dataset (with existing client assignments) to measure different metrics (*e.g.,* three-way-split).

separate data generating process. For example, for Federated EMNIST (Cohen et al., 2017), different clients write characters using different handwriting. Label-based partitioning does not account for this form of heterogeneity. To show this, in Figure 3 we visualize the clustering of client data between natural and label-based partitioning (Hsu et al., 2019) for Federated EMNIST. We project clients from each partitioning into a 2D space using T-SNE (Van der Maaten & Hinton, 2008) applied to the raw pixel data. Naturally partitioned examples clearly cluster more than label-based partitioned examples, which appear to be distributed similarly to the full data distribution.

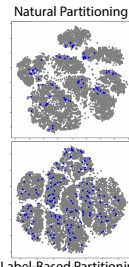

Natural Partitioning

Label-Based Partitioning

Figure 3: **T-SNE projection of different partitionings of EMNIST.** The top panel shows the naturally-partitioned dataset (partitioned by writer), the bottom panel shows the label-based synthetic dataset. The gray points are the projections of examples from each dataset, obtained by aggregating the data from 100 clients each. The blue points show projections of data from a single client. The naturally-partitioned client data appears much more tightly clustered, whereas the label-based partitioned data appears similarly distributed as the overall dataset, indicating that label-based partitioning may not fully represent realistic client heterogeneity.

Interestingly, differences in heterogeneity also significantly affect generalization behavior. In Figure 4, we compare the training progress of the naturally-partitioned EMNIST dataset with a label-based partitioning following the scheme by Hsu et al. (2019). Despite showing greater label heterogeneity (Fig. 4(a)), the label-based partitioning does not recover any significant participation gap, in sharp contrast to the natural partitioning (Fig. 4(d)). In Figure 5, we also see minimal participation gap in label-based partitioning for CIFAR. This motivates a client partitioning approach that better preserves the generalization behavior of naturally-partitioned datasets.

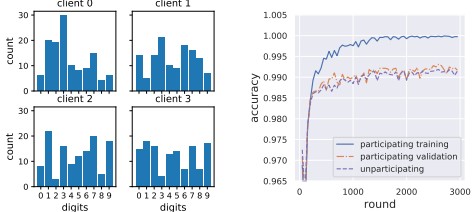 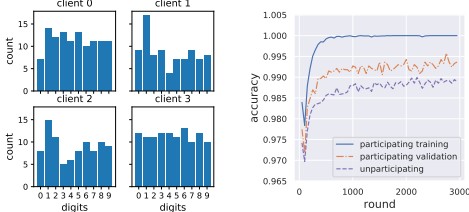

(a) Label histogram of label-based partitioning   (b) Learning progress of label-based partitioning   (c) Label histogram of natural partitioning   (d) Learning progress of natural partitioning

Figure 4: **Comparison of label-based synthetic partitioning and natural partitioning of EMNIST-10.** Observe that label-based partitioning shows greater label heterogeneity (a) than natural partitioning (c), while the participation gap (part_val − unpart) for label-based synthetic partitioning (b) is significantly smaller than that for the natural partitioning (d).

## 4.1 SEMANTIC CLIENT PARTITIONING AND THE PARTICIPATION GAP

To explore and remediate differences in client heterogeneity across natural and synthetic datasets, we propose a semantics-based framework to assign semantically similar examples to clients during federated dataset partitioning. We instantiate this framework via an example of an image classification task.

Our goal is to reverse-engineer the federated dataset-generating process described in Equation (1) so that each client possesses semantically similar data. For example, for the EMNIST dataset, we expect every client (writer) to (i) write in a consistent style for each digit (**intra-client intra-label similarity**) and (ii) use a consistent writing style across all digits (**intra-client inter-label similarity**). A simple approach might be to cluster similar examples together and sample client data from clusters. However, if one directly clusters the entire dataset, the resulting clusters may end up largely correlated to labels. To disentangle the effect of label heterogeneity and semantic heterogeneity, we propose the following algorithm to enforce intra-client intra-label similarity and intra-client inter-label similarity in two separate stages.

- Stage 1: For each label, we embed examples using a pretrained neural network (extracting semantic features), and fit a Gaussian Mixture Model to cluster pretrained embeddings into groups. Note that this results in multiple groups per label. This stage enforces intra-client intra-label consistency.
- Stage 2: To package the clusters from different labels into clients, we aim to compute an optimal multi-partite matching with cost-matrix defined by KL-divergence between the Gaussian clusters. To reduce complexity, we heuristically solve the optimal multi-partite matching by progressively solving the optimal *bipartite* matching at each time for randomly-chosen label pairs. This stage enforces intra-client inter-label consistency.

We relegate the detailed setup to Appendix D. Using this procedure we can generate clients which have similar example semantics. We show in Figure 5 that this method of partitioning preserves the participation gap. In Appendix D, we visualize several examples of our semantic partitioning on various datasets, which can serve as benchmarks for future works.

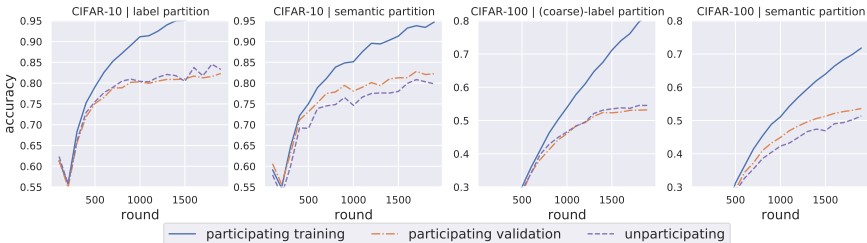

Figure 5: **Comparison of label-based partitioning and semantic partitioning (ours).** Results for CIFAR-10 and CIFAR-100 are shown. Observe that semantic partitioning recovers the participation gap typically observed in naturally-partitioned data.

## 4.2 EXPLAINING DIFFERENCES BETWEEN LABEL-BASED AND SEMANTIC PARTITIONING

To explain the above behavior, we revisit our mathematical setup and the definition of the participation gap. Recall that the participation gap is defined as (we omit the weights by setting $\rho_c \equiv 1$ for simplicity):

$$I_{\text{part\_gap}}(\boldsymbol{w}) := F_{\text{unpart}}(\boldsymbol{w}) - F_{\text{part\_val}}(\boldsymbol{w}) = \mathbb{E}_{c \sim \mathcal{P}}\left[\mathbb{E}_{\xi \sim \mathcal{D}_c}[f(\boldsymbol{w}; \xi)]\right] - \frac{1}{|\hat{\mathfrak{C}}|}\sum_{c \in \hat{\mathfrak{C}}}\left[\mathbb{E}_{\xi \sim \mathcal{D}_c}[f(\boldsymbol{w}; \xi)]\right] \quad (4)$$

In order to express the ideas without diving into details of measure theory, we assume without loss of generality that the meta-distribution $\mathcal{P}$ is a continuous distribution supported on $\mathfrak{C}$ with probability density function $p_{\mathcal{P}}(c)$. We also assume that for each client $c \in \mathfrak{C}$, the local distribution $\mathcal{D}_c$ is a continuous distribution supported on $\Xi$ with probability density function $p_{\mathcal{D}_c}(\xi)$. Therefore, the participation gap becomes

$$I_{\text{participation}}(\boldsymbol{w}) = \int_{\xi \in \Xi} f(\boldsymbol{w}; \xi) \cdot \left(\int_{c \in \mathfrak{C}} p_{\mathcal{D}_c}(\xi)p_{\mathcal{P}}(c)\mathrm{d}c - \frac{1}{|\hat{\mathfrak{C}}|}\sum_{c \in \hat{\mathfrak{C}}} p_{\mathcal{D}_c}(\xi)\right)\mathrm{d}\xi. \quad (5)$$

Therefore the scale of participation gap could depend (negatively) on the concentration speed from $\frac{1}{|\hat{\mathfrak{C}}|}\sum_{c \in \hat{\mathfrak{C}}} p_{\mathcal{D}_c}(\xi)$ to $\int_{c \in \mathfrak{C}} p_{\mathcal{D}_c}(\xi)p_{\mathcal{P}}(c)\mathrm{d}c$ as $|\hat{\mathfrak{C}}| \to \infty$.[3] We hypothesize that for label-based partitioning, the concentration is fast because each client has a large entropy as it can cover the entire distribution of a given label. On the other hand, for natural or semantic partitioning, the concentration is slower as the local distribution of each client has lower entropy due to the (natural or synthetic) semantic clustering.

We validate our hypothesis with an empirical estimation of local dataset entropy, shown in Figure 6. We observe that the clients generated by label-based partitioning demonstrate much higher entropy than the natural ones. Notably, our proposed semantic partitioning has a very similar entropy distribution across clients as the natural partitioning. This indicates that the heterogeneity in EMNIST is mostly attributed to semantic heterogeneity.

---

[3]One can make the above claim rigorous with standard learning theory approaches such as uniform convergence and Rademacher complexity (Vapnik, 1998).

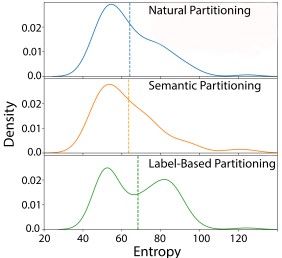

Figure 6: **Kernel density estimates of the distribution of client entropy for naturally-partitioned clients (top), semantic-partitioned clients (middle), and label-based partitioned clients (bottom).** While naturally and semantically partitioned clients appear to have approximately the same distribution of client entropies, the synthetically partitioned clients are distributed differently and have higher average entropy (48 Nats) than the other forms of partitioning (44 Nats). We refer readers to Appendix E for the detailed methodology for the estimation of the entropy.

Table 1: **Summary of experimental results.** We perform federated and centralized training across six tasks. EMNIST, Shakespeare, and StackOverflow are naturally-partitioned, and CIFAR is semantically partitioned. Observe that the participation gaps (gap between `part_val` and `unpart`) are consistent across tasks. We provide other metric statistics across clients (such as percentiles of metrics) in Table 2 of Appendix B.2.

|  | Federated Training | | | Centralized Training | | |
|---|---|---|---|---|---|---|
|  | `part_train` | `part_val` | `unpart` | `part_train` | `part_val` | `unpart` |
| EMNIST-10 | 100.0 | 99.6 | 98.9 | 100.0 | 99.5 | 98.9 |
| EMNIST-62 | 91.8 | 86.3 | 85.4 | 93.8 | 87.1 | 86.1 |
| CIFAR-10 | 97.5 | 83.3 | 81.6 | 99.7 | 86.3 | 84.9 |
| CIFAR-100 | 99.8 | 57.2 | 54.2 | 99.9 | 59.7 | 55.4 |
| Shakespeare | 58.8 | 56.5 | 56.2 | 60.7 | 57.2 | 56.8 |
| StackOverflow | 26.4 | 25.5 | 25.2 | 27.9 | 26.2 | 25.6 |

## 5 EXPERIMENTAL EVALUATION

We conduct experiments in six settings, including four image classification tasks: EMNIST-10 (digits only), EMNIST-62 (digits and characters) (Cohen et al., 2017; Caldas et al., 2019), CIFAR-10 and CIFAR-100 (Krizhevsky et al., 2009); and two next character/word prediction tasks: Shakespeare (Caldas et al., 2019) and StackOverflow (Reddi et al., 2021). We use FEDAVGM for image classification tasks and FEDADAM for text-based tasks (Reddi et al., 2021).[4] The detailed setups (including model, dataset preprocessing, hyperparameter tuning) are relegated to Appendix C. We summarize our main results in Figure 1 and Table 1. In the following subsections, we provide more detailed ablation studies exploring how various aspects of training affect generalization performance.

### 5.1 EFFECT OF THE NUMBER OF PARTICIPATING CLIENTS

In this subsection we study the effect of the number of participating clients on the generalization performance on various tasks. To this end, we randomly sample subsets of clients of different scales as participating clients, and perform federated training with the same settings otherwise. The results are shown in Figure 7. As the number of participating clients increases, the unparticipating accuracy monotonically improves, and the participation gap tends to decrease. This is consistent with our theoretical understanding, as the participating clients can be interpreted as a discretization of the overall client population distribution.

### 5.2 EFFECT OF CLIENT DIVERSITY

In this subsection, we study the effect of client diversity on generalization performance. Recall that in the previous subsection, we vary the number of participating clients while keeping the amount of training data per client unchanged. As a result, the total amount of training data will grow proportionally with the number of participating clients.

To disentangle the effect of diversity and the growth of training data size, in the following experiment, we instead fix the total amount of the training data, while varying the concentration across clients. The

---

[4]In addition, we experimented with FEDYOGI on these tasks. The performance is comparable (in terms of both participating validation and unparticipating metrics). We also experimented with vanilla FEDAVG and FEDADAGRAD, which are less effective than the other adaptive optimizers in these settings, but the participation gaps are generally consistent.

Figure 7: **Effect of the number of participating clients.** See Section 5.1 for discussion.

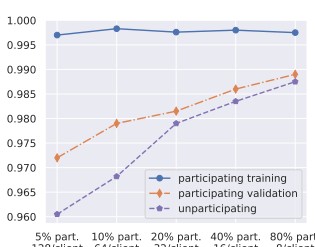

Figure 8: **Effect of diversity on generalization.** We fix the total amount of training data while varying the concentration across clients. The concentration varies from taking only 5% clients as participating clients where each client contributes 128 training data, to the most diverse distribution with 80% clients as participating clients but each client only contributes 8 training data. Observe that while the total amount of training data is identical, the more diverse settings exhibit better performance in terms of both participating validation and unparticipating accuracy.

experiment is conducted on the EMNIST digits dataset. As shown in Figure 8, the training data from a new participating client can be more valuable than those contributed by the existing participating clients. The intuition can also be justified by our model in that the data from a new participating client is drawn from the overall population distribution $\int_{c \in \mathfrak{C}} p_{\mathcal{P}}(c) p_{\mathcal{D}_c}(\xi) \mathrm{d}c$, whereas the data from existing clients are drawn from the distribution aggregated by existing clients $\frac{1}{|\hat{\mathfrak{C}}|} \sum_{c \in \hat{\mathfrak{C}}} p_{\mathcal{D}_c}(\xi)$. This reveals the importance of client diversity in federated training.

## 5.3 OVERVIEW OF ADDITIONAL EXPERIMENTS IN APPENDICES

We provide more detailed analysis and further experiments in Appendices B and C, including:

- Training progress of centralized optimizers on six tasks, see Appendix B.1.
- Detailed analysis of metrics distributions across clients, see Appendices B.2 and B.3. We observe that the unparticipating clients tend to exhibit longer tails on the lower side of accuracy.
- Results on alternative hyperparameter choices, see Appendices C.1.1 and C.2.1.
- Effect of multiple local epochs per communication round, see Appendix C.1.2.
- Effect of the strength of weight decay, see Appendix C.2.2.
- Effect of model depth, see Appendix C.2.3.

## 6 COMMUNITY SUGGESTIONS

In this work we have used the three-way-split, dataset partitioning strategies, and distributions of metrics to systematically study generalization behavior in FL. Our results inform the following suggestions for the FL community:

- Researchers can use the three-way split to disentangle out-of-sample and participation gaps in empirical studies of FL algorithms.
- When proposing new federated algorithms, researchers might prefer using naturally-partitioned or semantically-partitioned datasets for more realistic simulations of generalization behavior.
- Distributions of metrics across clients (*e.g.,* percentiles, variance) may vary across groups in the three-way split (see Table 2 and Figure 10). We suggest researchers report the distribution of metrics across clients, instead of just the average, when reporting metrics for participating and non-participating clients. We encourage researchers to pay attention to the difference of two distributions (participating validation and unparticipating) as it may have fairness implications.

## ACKNOWLEDGEMENTS

We would like to thank Zachary Charles, Zachary Garrett, Zheng Xu, Keith Rush, Hang Qi, Brendan McMahan, Josh Dillon, and Sushant Prakash for helpful discussions at various stages of this work.

## REPRODUCIBILITY STATEMENT

We provide complete descriptions of experimental setups, including dataset preparation and preprocessing, model configurations, and hyperparameter tuning in Appendix C. Appendix D describes the detailed procedure for semantic partitioning, and Appendix E presents the detailed approach for estimating the entropy that generates Figure 6.

We are also releasing an extensible code framework for measuring out-of-sample and participation gaps and distributions of metrics (*e.g.,* percentiles) for federated algorithms across several tasks.[5] We include all tasks reported in this work; the framework is easily extended with additional tasks. We also include libraries for performing label-based and semantic dataset partitioning (enabling new benchmark datasets for future works, see Appendix D). This framework enables easy reproduction of our results and facilitates future work. The framework is implemented using TensorFlow Federated (Ingerman & Ostrowski, 2019). The code is released under Apache License 2.0. We hope that the release of this code encourages researchers to take up our suggestions presented in Section 6.

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

# Appendices

LIST OF APPENDICES

# A  ADDITIONAL RELATED WORK

Recent years have observed a booming interest in various aspects of Federated Learning, including communication-efficient learning (McMahan et al., 2017; Konečný et al., 2016; Zhou & Cong, 2018; Haddadpour et al., 2019a; Wang & Joshi, 2018; Yu & Jin, 2019; Yu et al., 2019; Basu et al., 2019; Stich, 2019; Khaled et al., 2020; Yuan & Ma, 2020; Woodworth et al., 2020; Yuan et al., 2021; Li et al., 2021; Huang et al., 2021; Glasgow et al., 2021), model ensembling (Bistritz et al., 2020; He et al., 2020; Lin et al., 2020; Chen & Chao, 2021), integration with compression (Faghri et al., 2020; Gorbunov et al., 2020; Sohn et al., 2020; Beznosikov et al., 2020; Horváth & Richtárik, 2020; Albasyoni et al., 2020; Jiang et al., 2020; Islamov et al., 2021), systems heterogeneity (Smith et al., 2017; Diao et al., 2020), data (distributional) heterogeneity (Haddadpour et al., 2019b; Khaled et al., 2020; Li et al., 2020d; Koloskova et al., 2020; Woodworth et al., 2020; Mohri et al., 2019; Zhang et al., 2020; Li et al., 2020b; Wang et al., 2020a; Karimireddy et al., 2020; Pathak & Wainwright, 2020; Al-Shedivat et al., 2021), fairness (Wang et al., 2020b; Li et al., 2020c; Mohri et al., 2019), personalization (Smith et al., 2017; Nichol et al., 2018; Khodak et al., 2019; Balcan et al., 2019; Jiang et al., 2019; Wang et al., 2019; Chen et al., 2019; Fallah et al., 2020; Hanzely et al., 2020; London, 2020; T. Dinh et al., 2020; Yu et al., 2020; Hanzely & Richtárik, 2020; Agarwal et al., 2020; Deng et al., 2020; Hao et al., 2020; Liang et al., 2020), and privacy (Balle et al., 2020; Chen et al., 2020; Geiping et al., 2020; London, 2020; So et al., 2020; Zhu et al., 2020; Brown et al., 2020). These works often study generalization and convergence for newly proposed algorithms. Huang et al. (2021) studied the generalization of Federated Learning in Neural-tangent kernel regime. But, to our knowledge, there is no existing work that disentangles out-of-sample and participation gaps in federated training. We refer readers to (Kairouz et al., 2019; Wang et al., 2021) for a more comprehensive survey on the recent progress in Federated Learning.

# B  ADDITIONAL EXPERIMENTAL RESULTS

In this section, we present several experimental results omitted from the main body due to space constraints. Additional task-specific ablation experiments can be found in Appendix C.

## B.1  TRAINING PROGRESS OF CENTRALIZED OPTIMIZERS

In this subsection, we repeat the experiment in Figure 1 with centralized training. The results are shown in Figure 9. Observe that participation gap still exists with centralized optimizers. This is because the participation gap is an intrinsic outcome of the heterogeneity of federated dataset.

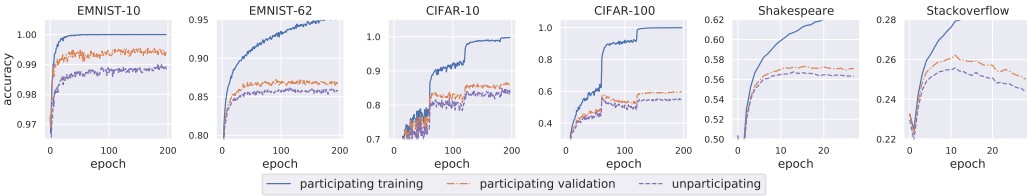

Figure 9: **Centralized training progress on six different federated tasks.** Observe that the participation gap still exists even with centralized optimizers. We refer readers to Table 1 for a quantitative comparison between federated optimizers and centralized optimizers.

## B.2  PERCENTILES OF METRICS ACROSS CLIENTS

In this subsection, we report the detailed statistics of metrics across clients. Recall that in Table 1, we aggregated the metrics across clients by weighted averaging, where the weights are determined by the number of elements contributed by each client. In the following Table 2, we report five percentiles of metrics across clients: 95th, 75th, 50th (a.k.a. median), 25th, and 5th. These statistics provide a detailed characterization on the metrics distribution across clients.[6]

---

[6]To make the percentiles comparable, we ensure the un-participating clients and participating validation clients have the same scale of elements per client.

Table 2: **Percentiles of metrics across clients on six federated tasks.** We observe that the unpartici-pating clients tend to exhibit longer tails on the lower side of accuracy. For example, the participating clients of EMNIST-10 have perfect (100%) accuracy even for clients at the 5th percentile, whereas the unparticipating clients only achieve 91.7%.

| | percentile | Federated Training | | Centralized Training | |
|---|---|---|---|---|---|
| | | part_val | unpart | part_val | unpart |
| EMNIST-10 | 95th | 100.0 | 100.0 | 100.0 | 100.0 |
| | 75th | 100.0 | 100.0 | 100.0 | 100.0 |
| | 50th | 100.0 | 100.0 | 100.0 | 100.0 |
| | 25th | 100.0 | 100.0 | 100.0 | 100.0 |
| | 5th | 100.0 | 91.7 | 100.0 | 91.7 |
| EMNIST-62 | 95th | 100.0 | 100.0 | 100.0 | 100.0 |
| | 75th | 93.3 | 92.3 | 93.5 | 93.5 |
| | 50th | 88.2 | 87.1 | 87.5 | 87.5 |
| | 25th | 82.1 | 78.6 | 81.8 | 79.2 |
| | 5th | 66.7 | 64.7 | 71.4 | 70.6 |
| CIFAR-10 | 95th | 93.2 | 90.0 | 95.5 | 92.9 |
| | 75th | 88.2 | 86.0 | 90.9 | 89.1 |
| | 50th | 83.0 | 81.3 | 87.0 | 85.7 |
| | 25th | 79.2 | 76.7 | 82.1 | 81.5 |
| | 5th | 71.1 | 69.6 | 77.1 | 73.7 |
| CIFAR-100 | 95th | 65.4 | 62.4 | 66.7 | 62.6 |
| | 75th | 61.1 | 56.7 | 64.2 | 56.6 |
| | 50th | 57.3 | 53.8 | 61.3 | 55.7 |
| | 25th | 53.9 | 51.9 | 55.5 | 52.8 |
| | 5th | 46.9 | 46.4 | 50.4 | 50.4 |
| Shakespeare | 95th | 68.4 | 71.4 | 70.1 | 71.4 |
| | 75th | 60.8 | 60.0 | 61.7 | 61.1 |
| | 50th | 57.3 | 56.8 | 58.2 | 57.5 |
| | 25th | 54.0 | 53.1 | 54.5 | 53.7 |
| | 5th | 38.4 | 38.2 | 42.6 | 40.9 |
| StackOverflow | 95th | 31.0 | 31.3 | 31.8 | 31.4 |
| | 75th | 27.7 | 27.7 | 28.8 | 27.9 |
| | 50th | 25.6 | 25.4 | 26.3 | 25.9 |
| | 25th | 23.5 | 23.2 | 24.2 | 23.7 |
| | 5th | 20.1 | 20.0 | 20.8 | 20.7 |

### B.3 FEDERATED TRAINING PROGRESS AT THE 25TH PERCENTILE ACORSS CLIENTS

To further inspect the distribution of metrics across clients, we plot the 25th percentile of accuracy across clients versus communication rounds (training progress). The results are shown in Figure 10.

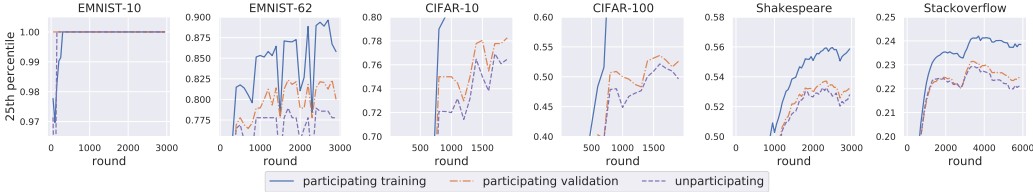

Figure 10: **Accuracies of the client at the 25th percentile versus the communication rounds.**

## C  ADDITIONAL DETAILS ON EXPERIMENTAL SETUP AND TASK-SPECIFIC EXPERIMENTS

In this section we provide details of the experimental setup, including dataset preparation/preprocessing, model choice and hyperparameter tuning. We also include task-specific experiments with ablations.

For every setting, unless otherwise stated, we tune the learning rate(s) to achieve the best sum of participating validation accuracy and unparticipating accuracy (so that the result will not be biased towards one of the accuracies).

### C.1  EMNIST HAND-WRITTEN CHARACTER RECOGNITION TASK

**Federated Dataset Description and Proprocessing.** The EMNIST dataset (Cohen et al., 2017) is a hand-written character recognition dataset derived from the NIST Special Database 19 (Grother & Flanagan, 1995). We used the Federated version of EMNIST (Caldas et al., 2019) dataset, which is partitioned based on the writer identification. We consider both the full version (62 classes) as well as the numbers-only version (10 classes). We adopt the federated EMNIST hosted by Tensorflow Federated (TFF). In TFF, federated EMNIST has a default intra-client split, namely all the clients appeared in both the "training" and "validation" dataset. To construct a three-way split, we hold out 20% of the total clients as unparticipating clients. Within each participating client, we keep the original training/validation split, *i.e.,* the original training data that are assigned to participating clients will become participating training data. We tested the performance under various number of participating clients, as shown in Figure 7. The results reported in Table 1 are for the case with 272 participating clients.

**Model, Optimizer, and Hyperparameters.** We train a shallow convolutional neural network with approximately one million trainable parameters as in (Reddi et al., 2021). For centralized training, we run 200 epochs of SGD with momentum = 0.9 with constant learning rate with batch size 50. The (centralized) learning rate is tuned from $\{10^{-2.5}, 10^{-2}, \dots, 10^{-0.5}\}$. For federated training, we run 3000 rounds of FEDAVGM (Reddi et al., 2021) with server momentum = 0.9 and constant server and client learning rates. For each communication round, we uniformly sample 20 clients to train for 1 epoch with client batch size 20. The client and server learning rates are both tuned from $\{10^{-2}, 10^{-1.5}, \dots, 1\}$.

### C.1.1  CONSISTENCY ACROSS VARIOUS HYPERPARAMETERS CHOICES

In Table 1, we only presented the best hyperparameter choice (learning rate combinations). In this subsubsection, we show that the pattern of generalization gap is consistent across various hyperparameter choices. The result is shown in Figure 11.

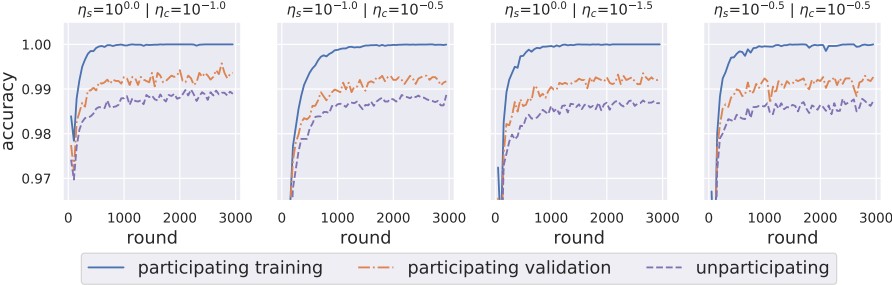

Figure 11: **Consistency of participation gaps across hyperparameter choice (learning rates configuration).** We present the best four (4) combination of learning rates for federated training of EMNIST-10. Here $\eta_c$ stands for client learning rate, and $\eta_s$ stands for server learning rate. We observe that the participation gap is consistent across various configurations of learning rates.

### C.1.2 Effect of multiple local epochs per communication round

In the main experiments we by default let each sampled client run one local epoch every communication round. In this subsubsection, we evaluate the effect of multiple local epochs on the generalization performance. The result is shown in Figure 12.

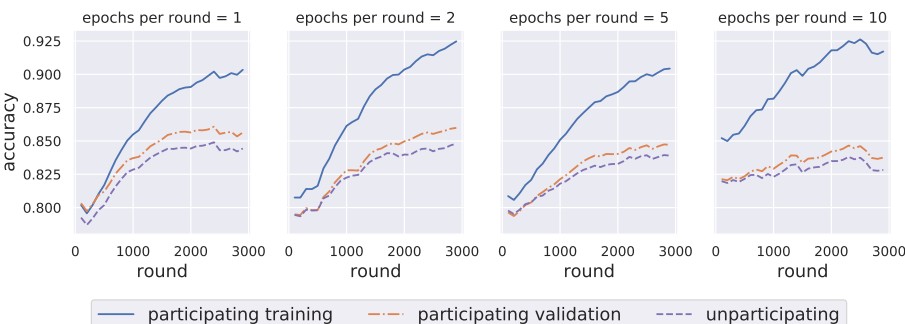

Figure 12: **Effect of multiple client epochs per round on EMNIST-62.** We repeat the experiment on EMNIST-62 but instead let each sampled client run multiple local epochs per communication round. The other settings (including the total communication rounds) remain the same. We observe that the participation gap is consistent across various settings of local epochs.

## C.2 CIFAR 10/100 Image Classification Task

**Federated Dataset Preprocessing.** The CIFAR-10 and CIFAR-100 datasets (Krizhevsky et al., 2009) are datasets of natural images distributed into 10 and 100 classes respectively. Since the dataset does not come with user assignment, we first shuffle the original dataset and assign to clients by applying our proposed semantic synthesized partitioning. The CIFAR-10 and CIFAR-100 dataset are partitioned into 300 and 100 clients, respectively. For three-way split, we hold out 20% (60 for CIFAR-10, 20 for CIFAR-100) clients as unparticipating clients, and leave the remaining client as participating clients. Within each participating client, we hold out 20% of data as (participating) validation data.

**Model, Optimizer, and Hyperparameters** We train a ResNet-18 (He et al., 2016) in which the batch normalization is replaced by group normalization (Wu & He, 2018) for improved stability in federated setting, as recommended by Hsieh et al. (2019). For centralized training, we run 200 epochs of SGD with momentum = 0.9 with batch size 50, and decay the learning rate by 5x every 60 epochs. The initial learning rate is tuned from $\{10^{-2.5}, 10^{-2}, \ldots, 10^{-0.5}\}$. For federated training, we run 2,000 rounds of FedAvgM (Reddi et al., 2021) with server momentum = 0.9, and decay the server learning rate by 5x every 600 communication rounds. For each communication round, we uniformly sample 10 clients (for CIFAR-100) or 30 clients (for CIFAR-10), and let each client train for 1 local epoch with batch size 20. The client learning rate is tuned from $\{10^{-2}, 10^{-1.5}, \ldots, 1\}$; the server learning rate is tuned from $\{10^{-1.5}, 10^{-1}, \ldots, 10^{0.5}\}$.

### C.2.1 Consistency across various hyperparameters choice

In the main result Table 1 we only present the best hyperparameter choice (learning rate combinations). In this subsubsection, we show that the pattern of generalization gap is consistent across hyperparameter choice. The result is shown in Figures 13 and 14.

### C.2.2 Effect of Weight Decay Strength

In the main experiments we by default set the weight decay of ResNet-18 to be $10^{-4}$. In this subsubsection, we experiment various other options of weight decay from $10^{-5}$ to $10^{-2}$

The result is shown in Figure 15.

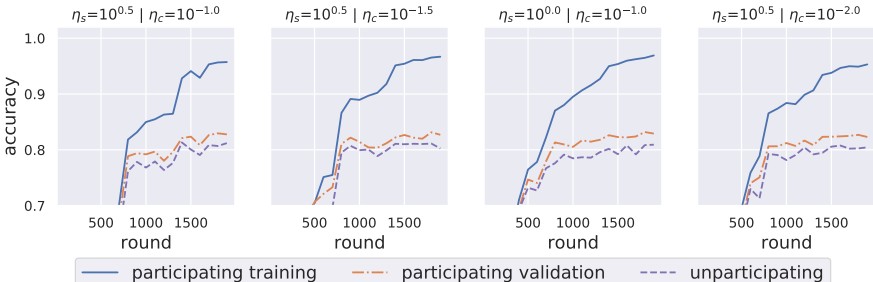

Figure 13: **Consistency of participation gaps across hyperparameter choice (learning rates configuration).** We present the best four (4) combination of learning rates for federated training of CIFAR-10. Here $\eta_c$ stands for client learning rate, and $\eta_s$ stands for server learning rate. We observe that the participation gap is largely consistent across various configurations of learning rates.

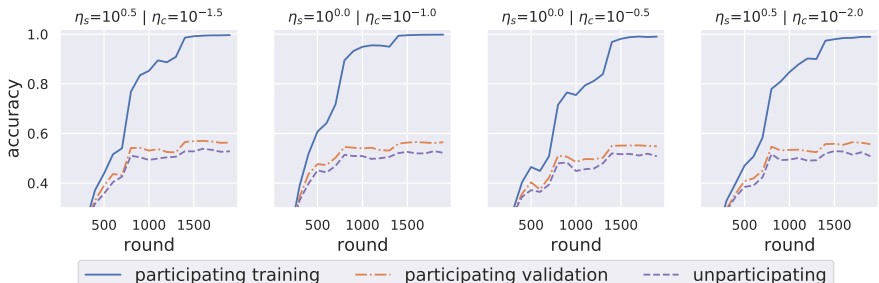

Figure 14: **Consistency of participation gaps across hyperparameter choice (learning rates configuration).** We present the best four (4) combination of learning rates for federated training of CIFAR-100. Here $\eta_c$ stands for client learning rate, and $\eta_s$ stands for server learning rate. We observe that the participation gap is consistent across various configurations of learning rates.

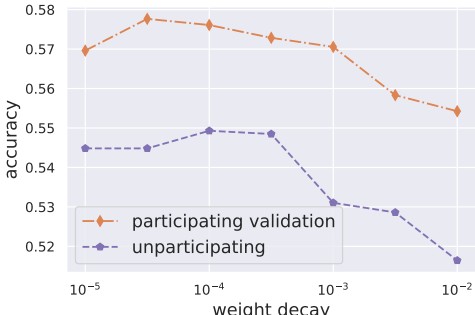

Figure 15: **Effect of $\ell_2$ weight decay on CIFAR-100 training.** We federated train the ResNet-18 networks for CIFAR-100 with various levels of weight decay ranging from $10^{-5}$ to $10^{-2}$. We observe that a moderate scale of weight decay might improve the unparticipating accuracy and therefore decrease the participation gap. However, an overlarge weight decay might hurt both participating validation and unparticipating performance.

### C.2.3 EFFECT OF MODEL DEPTH

In the main experiments we by default train a ResNet-18 for the CIFAR task. In this subsubsection, we experiment a deeper model (ResNet-50) for the CIFAR-100. The result is shown in Figure 16.

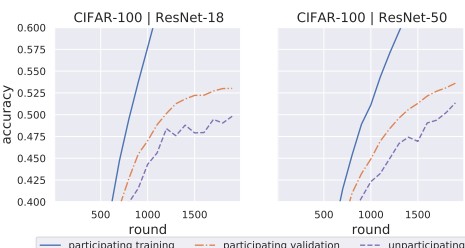

Figure 16: **Effect of a deeper ResNet on CIFAR-100 training.** We federatedly train a ResNet-50 for CIFAR-100 to compare with our default choice (ResNet-18). We apply a constant learning rate (instead of step decay learning rate) for easy comparison. We observe that while using a deeper model improves the overall accuracy, the participation gap is still reasonably large for ResNet-50.

### C.3 SHAKESPEARE NEXT CHARACTER PREDICTION TASK

**Federated Dataset Description and Preprocessing.** The Shakespeare dataset (Caldas et al., 2019) is a next character prediction dataset containing lines from *the Complete Works of William Shakespeare* where each client is a different character from one of the plays. We adopt the federated shakespeare dataset hosted by Tensorflow Federated (TFF). In TFF, the federated shakespeare dataset was by default split intra-cliently, namely all the clients appeared in both the "training" and "validation" dataset. To construct a three-way split, we hold out 20% of the total clients as unparticipating clients, and leave the remaining (80%) clients as participating clients (which gives the result reported in Table 1). Within each participating client, we keep the original training/validation split, e.g., the original training data that are assigned to these participating clients will become participating training data. We also tested the performance under other numbers of participating clients, as shown in Figure 7.

**Model, Optimizer, and Hyperparameters** We train the same recurrent neural network as in (Reddi et al., 2021). For centralized training, we run 30 epochs of Adam (with $\epsilon = 10^{-4}$) with batch size 20. We tune the centralized learning rate from $\{10^{-3}, 10^{-2.5}, \ldots, 10^{-1}\}$. For federated training, we run 3,000 rounds of FEDADAM (Reddi et al., 2021) with server $\epsilon = 10^{-4}$. For each communictaion round, we uniformly sample 10 clients, and let each client train for 1 local epoch with batch size 10. Both client and server learning rates are tuned from $\{10^{-2}, 10^{-1.5}, \ldots, 1\}$.

### C.4 STACKOVERFLOW NEXT WORD PREDICTION TASK

**Federated Dataset Description and Preprocessing.** The Stack Overflow dataset consists of questions and answers taken from the website Stack Overflow. Each client is a different user of the website. We adopt the stackoverflow dataset hosted by Tensorflow Federated (TFF). In TFF, the federated stackoverflow dataset is splitted **inter-cliently**, namely the training data and validation data belong to two disjoint subsets of clients. To construct a three-way split, we will treat the original "validation" clients as unparticipating clients. Within each participating client, we randomly hold out the max of 20% or 1000 elements as (participating) validation data, and the max of 80% or 1000 elements as (participating) training data. Due to the abundance of stackoverflow data, we randomly sample a subset of clients from the original "training" clients as participating clients. The result shown in Table 1 is for the case with 3425 participating clients. We also tested other various levels of participating clients, shown in Figure 7.

**Model, Optimizer, and Hyperparameters** We train the same recurrent neural network as in (Reddi et al., 2021). For centralized training, we run 30 epochs of Adam (with $\epsilon = 10^{-4}$) with batch size 200. We tune the centralized learning rate from $\{10^{-3}, 10^{-2.5}, \ldots, 10^{-1.5}\}$. For federated training, we run 6,000 rounds of FEDADAM (Reddi et al., 2021) with server $\epsilon = 10^{-4}$. For each communictaion round, we randomly sample 100 clients, and let each client train for 1 local epoch with batch size 50. Both client and server learning rates are tuned from $\{10^{-2}, 10^{-1.5}, \ldots, 1\}$. The client learning rate is tuned from $\{10^{-3}, 10^{-1.5}, \ldots, 10^{-1}\}$; the server learning rate is tuned from $\{10^{-2}, 10^{-1.5}, \ldots, 1\}$.

# D    DETAILS OF SEMANTICICALLY PARTITIONED FEDERATED DATASET

## D.1    DETAILS OF THE SEMANTIC PARTITIONING SCHEME

In this section we provide the details of the proposed algorithm to semantically partition a federated dataset for CIFAR-10 and CIFAR-100. For clarify, we use $K$ to denote the number of classes, and $C$ to denote the number of clients partitioned into.

The first stage aims to cluster each label into $C$ clusters.

1. Embed the original inputs of dataset using a pretrained EfficientNetB3. This gives a embedding of dimension 1280 for each input.
2. Reduce the dimension of the above embeddings to 256 dimensions via PCA.[7]
3. For each label, fit the corresponding input with a Gaussian mixture model with $C$ clusters. This step yields $C$ gaussian distribution for each of the $K$ labels. Formally, we let $\mathcal{D}_c^k$ denote the (Gaussian) distribution of the cluster $c$ of label $k$.

The second stage will package the clusters from different labels across clients. We aim to compute an optimal multi-partite matching with cost-matrix defined by KL-divergence between the Gaussian clusters. To reduce complexity, we heuristically solve the optimal multi-partite matching by progressively solving the optimal bipartite matching at each time for some randomly-chosen label pairs. Formally, we run the following procedure

---
1: Initialize $S_{\text{unmatched}} \leftarrow \{1, \ldots, K\}$
2: Randomly sample a label $k$ from $S_{\text{unmatched}}$, and remove $k$ from $S_{\text{unmatched}}$.
3: **while** $S_{\text{unmatched}} \neq \emptyset$ **do**
4:     Randomly sample a label $k'$ from $S_{\text{unmatched}}$, and remove $k'$ from $S_{\text{unmatched}}$.
5:     Compute a cost matrix $\boldsymbol{A}$ of dimension $C \times C$, where $\boldsymbol{A}_{ij} \leftarrow D_{\text{KL}}(\mathcal{D}_i^k || \mathcal{D}_j^{k'})$.
6:     Solve and record the optimal bipartite matching with cost matrix $\boldsymbol{A}$.
7:     Set $k \leftarrow k'$
8: **return**  the aggregation of all the bipartite matchings computed.

---

## D.2    VISUALIZATION OF SEMANTICICALLY PARTITIONED CIFAR-100 DATASET

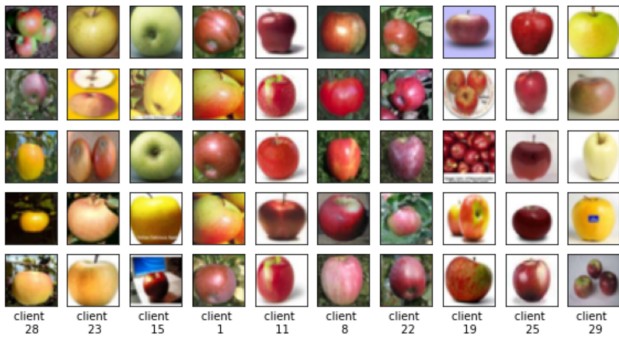

Figure 17: **Visualization of semantic partitioning of CIFAR-100.** We partition the CIFAR-100 dataset into 100 clients without resorting to external user information (such as writer identification). Here we show 10 out of 100 clients featuring the label "apple".

## D.3    VISUALIZATION OF SEMANTICALLY PARTITIONED MNIST DATASET

---

[7]Reducing the dimension is purely a computational issue since the original embedding dimension (1280) is too large for downstream procedures such as GMM fitting and optimal matching (measured by KL divergence). While there may be other complicated dimension reduction technique, we found PCA to be simple enough to generate reasonable results. The dimension of 256 is a trade-off of (down-stream) computational complexity and embedding information.

Figure 18: **Visualization of semantic partitioning of MNIST.** We partition the (classic) MNIST dataset into 300 clients without resorting to external user information (such as writer identification). Here we show 5 out of 300 clients. Observe that the images within each client demonstrates consistent writing styles both within label and across labels.

# E    METHODOLOGY FOR COMPUTING ENTROPY

We hypothesize that a participation gap exists for naturally partitioned datasets and not for synthetically partitioned datasets because the naturally partitioned datasets inherently contain correlated inputs not drawn IID from the full data generating distribution. Put another way, the entropy of the input data for a given label from a naturally partitioned client is lower than the entropy for that same label from a synthetically partitioned client. To evaluate this claim, we need to (approximately) infer the data generating distribution for each client, and then measure the entropy of this distribution, defined as:

$$H(q) = -\mathbb{E}_{x \sim q(x)} \log q(x) \tag{6}$$

To infer the client data generating distribution, we used deep generative models. Because our clients possess relatively few training examples ($\mathcal{O}(10)$ for a particular class), many deep generative models such as Glow (Kingma & Dhariwal, 2018) or PixelCNN (Salimans et al., 2017) will not be able to learn a reasonable density model. We instead used a Variational Autoencoder (Kingma & Welling, 2013) to approximate the deep generative process. This model is significantly easier to train compared to the much larger generative models, but does not have tractable log-evidence measurement. Instead, models are trained by minimizing the negative Evidence Lower Bound (ELBO).

We filtered each client to contain data only for a single label. Because of the sparseness of the data after filtering, we found that a 2 dimensional latent space was sufficient to compress our data without significant losses. We used a Multivariate Normal distribution for our posterior and prior, and an Independent Bernoulli distribution for our likelihood. The posterior was given a full covariance matrix to account for correlations in the latent variable. All models were trained for $10^4$ training steps.

In order to evaluate our models, we used a stochastic approximation to the log-evidence, given by a 1000 sample IWAE (Burda et al., 2015). IWAE is a lower bound on the Bayesian Evidence that becomes asymptotically tight when computed with a large number of samples. We evaluated the entropy for 100 clients from naturally partitioned, syntactically partitioned, and synthetically partitioned datasets, and computed the average across clients as our estimate for the client data entropy. We find that synthetic partitioning results in an average client entropy of 50 Nats, while Natural partitioning results in clients with only 40 Nats of entropy. Syntactic partitioning falls in between these two, having 45 Nats of entropy.

