# OpenReview forum: "What Do We Mean by Generalization in Federated Learning?"
_ICLR.cc/2022/Conference — ICLR 2022 Poster_

### Official Review · Reviewer_dKHn · 2021-10-30

**Correctness:** 3
**Technical Novelty And Significance:** 2
**Empirical Novelty And Significance:** 2
**Recommendation:** 6
**Confidence:** 3

**Main Review:**

Decomposing the performance gap into out-of-sample gap and participation gap is not new. The very simple way is to split the whole dataset into training and validation sets, and the training data is distributed across workers, then training data on each worker is further split into local training and local validation data. In this way, the local validation loss could be viewed as the out-of-sample gap and the global validation loss can be viewed as the participation gap. Evaluating the loss on local data can be found in [1,2].

Another question is if some clients never participate in training, how can we evaluate the participation gap since the loss should be calculated on these data?

The experimental results are trivial from the view of the commonly-used data splitting described in the first paragraph. Could the authors please elaborate on the novelty of this paper? How can we deal with the performance gap due to data heterogeneity?


[1] Liang, Paul Pu, et al. "Think locally, act globally: Federated learning with local and global representations." arXiv preprint arXiv:2001.01523 (2020).
[2] Hao, Weituo, et al. "WAFFLe: Weight Anonymized Factorization for Federated Learning." arXiv preprint arXiv:2008.05687 (2020).

-------------------
After reading all the reviews and the rebuttal, some of my concerns have been addressed, I am willing to increase the grade.

**Summary Of The Paper:**

The paper proposes a framework for disentangling the performance gap in federated learning into out-of-sample gap and participation gap, which should serve as a better tool for explaining the model generalization performance. They also give a semantic synthesis strategy that enables realistic simulation to create more natural federated data splitting. More analysis on the relationship between gaps and data heterogeneity is provided to support the validity of the proposed splitting framework.

**Summary Of The Review:**

I feel the idea and the experimental results are trivial if we view it from a commonly-used federated data splitting perspective. If the authors can propose some remedy to this participation gap, this will be an interesting piece of paper.

---

> ### Author Response · Authors · 2021-11-13
> **Author Response to Reviewer dKHn | Part 1 of 2**
>
> We thank the reviewer for their time and for the pointers to related work. In our updated manuscript, we have incorporated these papers into our related work section (see §A). We did want to clarify the significance of our work and relationship to other work. We're happy to answer any follow-up questions!
>
> We noticed there is a major misunderstanding in the reviewer's comment regarding the three-way split and participation gap, which we quote below:
> > ***Decomposing the performance gap into out-of-sample gap and participation gap is not new. The very simple way is to split the whole dataset into training and validation sets, and the training data is distributed across workers, then training data on each worker is further split into local training and local validation data. In this way, the local validation loss could be viewed as the out-of-sample gap and the global validation loss can be viewed as the participation gap. Evaluating the loss on local data can be found in [1,2] ... Could the authors please elaborate on the novelty of this paper?***
>
> We respectfully disagree with the claim that either [1] or [2] (or any other works to the best of our knowledge) proposed the three-way split or defined the participation gap as we proposed. Most importantly, the gaps observed in [1, 2] are **not** the participation gap (performance gap from whether a client contributes data to shared model), but are instead caused by a difference in models (gap from whether models are adapted to local data).
>
> To make this point clearer, we reproduce the following table from Table 1 of [1]:
>
> | Data Method | Local Test Acc. | New Test Acc.  | ... |
> |-------------|:---------------:|----------------|-----|
> | FedAvg      |   58.99 ± 1.50  |  58.99 ± 1.50  | ... |
> | Local only  |   87.93 ± 2.14  |  10.03 ± 0.06  | ... |
> | MTL         |   89.68 ± 0.75  |  10.06 ± 0.11  | ... |
> | LG-FedAvg   |   91.07 ± 0.50  |  57.95 ± 1.48  | ... |
> | LG-FedAvg   |   91.77 ± 0.56  |  60.79 ± 1.45  | ... |
>
> The "Local Test Acc." corresponds to evaluating a **personalized** model (e.g., for LG-FedAvg, a hybrid local and global model) for each particular client. The "New Test Acc." cooresponds to evaluating an **unpersonalized** model on an unseen client. The large gaps observed for the last 4 rows are caused by the usage of different models (**personalized** vs **unpersonalized**), and the participation gap (measured using the same model) cannot be disentangled from this gap. This is important, since proposing a framework to disentangle the participation gap from other generalization gaps is one of the main contributions of our work.
>
> - As a side note, note that the "Local Test Acc." and "New Test Acc." are identical with FedAvg method. This is another evidence of the difference of their splits and our splits --- if they were applying our proposed three-way split, **there would be a difference between these two columns**.
>
>
> Finally, we would like to highlight (as discussed in §1.1 of the original manuscript) that our work is in fact **orthogonal** and complementary to personalization works. In a personalized setting where unparticipating clients can also be personalized (e.g., fine-tuning), the participation gap can be defined by the difference between *personalized participating performance* and *personalized unparticipating* performance.
>
>
> || Participating (contributing training data)  | Un-participating (not contributing training data)|
> |-------------|-------------|:---------------:|
> |**Personalized (adapted to local data)**| Personalized Participating (I)  | Personalized Un-participating (II) |
> |**Un-personalized (not adapted to local data)**| Un-personalized Participating (III) | Un-personalized Un-participating (IV)   |
>
> To summarize, our work proposed the three-way split to study the  "participation gap" defined by (III)-(IV). [1, 2] and other personalization works study the "personalization gap" defined by (I) - (III) or (I) - (IV). We could also define participation gap in personalized settings as (I)-(II), which is an interesting future direction.

---

> ### Author Response · Authors · 2021-11-13
> **Author Response to Reviewer dKHn | Part 2 of 2**
>
> > ***Q: Another question is if some clients never participate in training, how can we evaluate the participation gap since the loss should be calculated on these data?***
>
> We thank the reviewer for this question, since this is exactly the main motivation of three-way split: if we want to evaluate the participation gap, we have to hold out some contributing clients as "unparticipating". These clients will evaluate the model with their local data and compute metrics, but will not actively impact the model by contributing training data.
>
> As discussed in §3, this scenario is akin to the motivation for the classic Expected Risk Minimization (ERM) framework in ML: if one wants to predict the performance on unseen data, one has to hold out some data as validation/test data. In FL, in order to predict the performance on un-participating clients, one has to hold out some clients as "validation" clients to evaluate. We refer to this group of clients as "un-participating" mainly to avoid overloading the terminology "validation", as there is another level of train/val split (training and validation data for participating clients).
>
> ---
>
> > ***Q: How can we deal with the performance gap due to data heterogeneity?***
>
> First, we would like to point out that the participation gap still persists even with centralized training (see Figure 9). This is different from the convergence efficiency issue also caused by heterogeneity, which can be fully resolved by training with centralized optimizers. Most of the FL literature tackling heterogeneity so far are devoted to this convergence issue, e.g., Adaptive Federated optimizers, FedProx, SCAFFOLD and FedDyn to name a few. In the extreme case, as long as these federated optimizers converge to the same point (which is the case for convex models and sufficiently small learning rate), they should experience the same participation gap. In our experiments, we indeed observe consistent participation gaps across various tested optimizers (FedAvg, FedAvgM, FedAdam, etc), which suggests a level of optimizer-agnosticity of the participation gap. We also extensively experiment with different hyperparameter choices, varying the computation/communication tradeoff, adding weight decay, and varying model architecture in §C. We generally observe a consistent participation gap across these variations.
>
> Therefore, the participation gap appears to be an intrinsic challenge when data comes from a distribution of distributions (as in FL), neglected in current literature in FL. Besides the ones we experimented with in the paper, we propose techniques that could potentially alleviate participation gap: 1) design novel client-level regularizers (we already observed that common regularizers do not typically reduce participation gap), 2) explicitly apply or adapt domain generalization techniques or robust learning techniques. These are obviously interesting future directions, but are far out of the scope of the current submission.
>
> ---
>
> Given these clarifications, particularly around core motivation of our framework for studying generalization, we hope the reviewer will consider adjusting their initial rating.

---

### Official Review · Reviewer_b9Rh · 2021-11-01

**Correctness:** 4
**Technical Novelty And Significance:** 4
**Empirical Novelty And Significance:** 4
**Recommendation:** 8
**Confidence:** 4

**Main Review:**

- **Pros**
    - The paper is well written and easy to understand for most parts.
    - To the best of my knowledge, existing works do not take into account the participation gap in their analysis or in their datasets split, therefore this work can be an important contribution to the FL community.
    - the authors provide all the necessary information about the results of the experiments, the hyperparameters tuning and the specifications to reproduce them, making this work a reference benchmark for future extensions.
    - the proposed framework is scalable and easily adaptable to the existing state-of-the-art strategies and methods.
    - assumptions and hypotheses are verified either theoretically or empirically.
    - release of an open-source framework for semantic-based dataset partitioning and participation gaps measurement.
- **Cons**
    - In Sec. 3, the authors propose the initial analysis on EMNIST-62 and then continue on CIFAR datasets, without explaining what happens to the previous setting (label-based or natural partitioning) when their contribution is introduced (semantic-based partitioning).
- **Questions**
    1. On results consistency to model size and hyperparams:
        - 1a) By looking at supp C1.1 and the consistency of results for different LR choices, I was wondering if the participation gaps are sensitive to the choice of the server optimizer.
        - 1b) Actually, more in general I think that the analysis would be much more complete if you included experiments with recent methods that have been developed to reduce heterogeneity issue in federate learning. eg. MIME, FedDyn, Scaffold, FedProx, FedBN,  etc.. It would be great to see if the participation gap is sensitive to the algorithm or optimization method.
        - 1c) Is the partecipation gap sensitive to the choice of the number of clients sampled at each communication round?
        - "1d) "the participation gap is still reasonably large for ResNet-50." Results in C2.3 figure 16 for resnet50 are not at final convergence. Is this claim valid at the end of training? Also, I imagine that these gaps should be independent on the choice of the architecture e.g efficientNet or others. Have you verified that?
        - 1e) very deep learner question: I'm curious to see how these gaps behave on large scale datasets as Inaturalist or Google Landmarks, but this is out of the scope of the paper.
    2. it would be interesting to see how the experiments in Fig 4 change when EMNIST is created with the semantic partitioning
    3. In the label-partitioned CIFAR, which is the chosen alpha for the Dirichlet’s distribution?
    4. Table 1: looking at the plots in Appendix B.1, the part_train accuracy on Shakespeare looks close to 0.62, where does the 60.7 in the table come from?
    5. How are the datasets balanced between clients (i.e. mean and stdev of data distributions among clients)?
    6. In Appendix C, you provide detailed information on the experimental setup. The best learning rates chosen for the experiments in Table 1 are not clear.
- **Minors**
    - Section 5.1, third row: typo “participating” is misspelled


**Summary Of The Paper:**

- The authors propose a study on generalization in Federated Learning (FL). They focus on the cross-device setting, where the clients are sampled from a large population and are characterized by unreliable availability, which implies that many clients may actually never take part in the training process. Driven by such realistic assumptions, they distinguish between participating and non-participating clients and identify two generalization gaps:
    - the __out-of-sample__ __gap__, i.e. the difference between the empirical and expected risk for participating clients, or the gap in performance of the model when tested on previously unseen data;
    - the __participation gap__, i.e. the difference in expected risk between the participating and non-participating clients, or the model drop in performance when faced with unseen distributions (i.e. new clients).  This is a direct consequence of the heterogeneity in FL: in heterogeneous settings, clients have access to different data distributions, leading the model to worse performance when new distributions are introduced. So the gap becomes a natural measurement for dataset heterogeneity, capability of the model to generalize and client diversity.
- The authors propose a three-way split for federated datasets in order to  perform an estimation of the risks and gaps: by identifying participating and non-participating clients first, and then training and validate data within the participating clients’ local data distribution.
- Finally, they also point out how federated datasets are usually split based on the labels distribution (each client is assigned a distribution on the labels), or on some natural information, e.g. the writer of a character, and show that this __label-based partitioning__ may not fully represent realistic client heterogeneity.
    - Therefore, they propose a semantics-based framework for partitioning the dataset, which behaves similarly to the natural and more realistic method, without requiring naturally-partitioned data.
- Their extensive analysis is tested on six different tasks and all information for experiments reproducibility are provided. Studies on the hyperparameters tuning are proposed as well.


**Summary Of The Review:**

A very nice paper. It proposes to measure both out-of-sample generalization and participation gaps and give several recommendations for training federated learning models following a new protocol. It also proposes a new way to create synthetic federated datasets that reflect heterogeneity in a more natural way than label-based split.  The experimental analysis is very concrete and the results look solid. I recommend acceptance.

---

> ### Author Response · Authors · 2021-11-13
> **Author Response to Reviewer b9Rh | Part 1 of 2**
>
> We thank the reviewer for their thorough review and valuable suggestions. We're glad the reviewer appreciated the novelty of the participation gap and three-way-split and their importance for the FL community, the extensive experimental results, and the open-source code framework release. We address each of the reviewer's comments:
>
> > ***Sec. 3, the authors propose the initial analysis on EMNIST-62 and then continue on CIFAR datasets, without explaining what happens to the previous setting (label-based or natural partitioning) when their contribution is introduced (semantic-based partitioning).***
> > ***2. It would be interesting to see how the experiments in Fig 4 change when EMNIST is created with the semantic partitioning***
>
> We would like to assure the reviewer that the same phenomenon (as in Figure 5) happens for EMNIST-10 dataset, i.e., semantic partitioning recovers participation gap. (This is not surprising though given the visual clustering in Figure 18.) We decided to present the CIFAR result at Figure 5 mostly because EMNIST-10 comes with a natural partitioning (by writers, results presented in Figure 1 showing a participation gap), while CIFAR does not, and we wanted to highlight our contribution on introducing an alternative (more natural) partitioning for CIFAR. We will add the EMNIST-10 version of this figure in the camera ready version.
>
> ---
>
> > ***1.On results consistency to model size and hyperparams:***
> > ***1a) By looking at supp C1.1 and the consistency of results for different LR choices, I was wondering if the participation gaps are sensitive to the choice of the server optimizer.***
> > ***1b) Actually, more in general I think that the analysis would be much more complete if you included experiments with recent methods that have been developed to reduce heterogeneity issue in federate learning. eg. MIME, FedDyn, Scaffold, FedProx, FedBN, etc.. It would be great to see if the participation gap is sensitive to the algorithm or optimization method.***
>
> We first remark on the sensitivity to optimizers. First, we notice that the participation gap still exists even with centralized optimizers (see Figure 9). This is different from the convergence efficiency issue, also caused by heterogeneity, which can be fully resolved by replacing with centralized optimizers. Most of the FL literature tackling heterogeneity so far are devoted to the convergence issue, e.g., Adaptive Federated optimizers, FedProx, SCAFFOLD, MIME, and FedDyn. In the extreme case, as long as these federated optimizers converge to the same point (which is the case for convex models and sufficient small learning rate), they should experience the same participation gap.
>
> Consequently, a priori we don't expect these federated optimizers to significantly improve the participation gap. In our experiments, we indeed observe consistent participation gaps across various tested optimizers (**FedAvg, FedAvgM, FedAdam, FedYogi, FedAdagrad, centralized, etc**), which suggests a level of optimizer-agnosticity of participation gap. As the reviewer mentioned, we believe a complete benchmark on other optimizers (e.g., MIME, FedDyn, etc) could be very beneficial to FL community. This is a great future direction, but it is out of the scope of the current manuscript.
>
> As a side note, we also extensively experiment with different hyperparameter choices, varying the computation/communication tradeoff, adding weight decay, and varying model architecture in Appendix C. We generally observe a consistent participation gap across these variations.
>
> ---
>
> > ***1c) Is the partecipation gap sensitive to the choice of the number of clients sampled at each communication round?***
>
> Our experiments suggest it is not sensitive to the choice of the number of clients sampled per round for a fixed population size, but we do see significant differences when we vary population size (see Figure 7 and Section 5.1 for discussion).

---

> ### Author Response · Authors · 2021-11-13
> **Author Response to Reviewer b9Rh | Part 2 of 2**
>
> > ***"1d) "the participation gap is still reasonably large for ResNet-50." Results in C2.3 figure 16 for resnet50 are not at final convergence. Is this claim valid at the end of training?***
>
> Yes, we can confirm. In fact, this figure is an artifact caused by learning rate tuning (recall that for each experiment we always tune the best server + client learning rate that achieves the best averaged participation validation and unparticipating accuracy). The ResNet-50 experiment happens to choose a smaller learning rate combination, which makes it converge slower. With the same number of rounds, we obtain similar accuracies (with similar gaps) by using a larger learning rate combination, which converges faster and plateaues earlier. The difference of these two figures are mostly caused by different choices of learning rates instead of the difference of layers.
>
> ---
>
> > ***Also, I imagine that these gaps should be independent on the choice of the architecture e.g efficientNet or others. Have you verified that?***
>
> Great question. For the pretrained embedding, we tested various versions of EfficientNet (B0, B3, B7), and pretrained ResNet. The conclusion is that as long as the pretrained network is powerful enough to generate useful features, then the participation gap will appear. For MNIST, all of the aforementioned networks can work. For CIFAR-100, I found EfficientB3 and EfficientB7 work better in generating semantic-partitioned clients (both qualitatively and quantitatively).
>
> ---
>
> > ***3. In the label-partitioned CIFAR, which is the chosen alpha for the Dirichlet’s distribution?***
>
> For CIFAR-10, the alpha is 100. (We also tested 10 and 1000, which give similar results). For CIFAR-100, we applied two levels of dirichlet distribution using the coarse label information. The coarse alpha is 10, the fine alpha is 1000. We will add these to our paper, thanks for the question.
>
> ---
>
> > ***4. Table 1: looking at the plots in Appendix B.1, the part_train accuracy on Shakespeare looks close to 0.62, where does the 60.7 in the table come from?***
>
> When generating the table, we pick the **single** epoch that attains the best averaged participating validation and unparticipating accuracy, and report all of the metrics **at that particular epoch**. (As a side note, this approach allows us to evaluate the gap on a consistent model.) For this Shakespeare experiment, the best averaged validation metric was attained at around the 10th epoch; at this time the training accuracy is around 60.7%. After that, the validation metrics start to decrease due to overfitting. Although the part_train accuracy reaches >62% eventually, those epochs were not recorded because the validation metrics are worse.
>
> ---
>
> > ***5. How are the datasets balanced between clients (i.e. mean and stdev of data distributions among clients)?***
>
> Please correct us if we are misunderstanding the question. If the reviewer was referring to the distribution of **metrics** across clients, please check out §B.2. If they were referring to the distribution of **dataset size** across clients, we intentionally make it well-balanced to disentangle from other effects. For example, for CIFAR-10 dataset with 300 synthetic clients, the mean dataset size is 200, the stddev is 43.
>
> ---
>
> > ***6. In Appendix C, you provide detailed information on the experimental setup. The best learning rates chosen for the experiments in Table 1 are not clear.***
>
> We thank the reviewer for this comment. We will definitely report these numbers in the camera-ready version.
>
> ---
>
> We thank the reviewer again for their valuable suggestions–they have significantly improved our work.

---

### Official Review · Reviewer_bfMQ · 2021-11-02

**Correctness:** 3
**Technical Novelty And Significance:** 2
**Empirical Novelty And Significance:** 2
**Recommendation:** 6
**Confidence:** 4

**Main Review:**

The paper points out an often overlooked issue in federated learning: when clients are sampled for averaging, some clients may hold deprecated models. Those clients will have a worse performance than their recently averaged fellows. The paper shows that for homogeneous local data distributions, this error vanishes in expectation, but remains an issue for heterogeneous distributions. These insights are valuable.

The main contributions of the paper, however, are fairly straight-forward and the theoretical result (Prop. 3.1) is trivial. The proposed three-way-split is a straight-forward estimator that gives little novel insights. The semantic partitioning of data into heterogeneous local datasets is interesting, but the advantage of the method over other strategies (simple distance based clustering, or using semantically similar datasets (e.g., MNIST, MNIST-M, SVHN, USPS, and SynthDigits)) remains unclear. The empirical evaluation is sound and confirms the effect of participating clients on the overall performance, but yield little novel insights.

Regarding the community suggestions, I feel that (at least in my bubble) these suggestions are not novel. Using naturally partitioned datasets is fairly common in personalized FL or FL from heterogeneous data sources (e.g., in FedBN [1]). The semantic partitioning might be an interesting new technique, but from this paper its benefits remain unclear. Suggesting to report the distribution of metrics seems particularly odd, since this is common practice in most ML papers (at least, reporting standard deviation or variance). In FL, the variance of local client performances is also often reported, or visually illustrated. Only the first suggestion - to analyze the participation gap - seems appropriate, but is limited to approaches where clients are sampled and - as the experiments suggest - the sampling probability for individual clients is sufficiently low.

Given this lack of novelty, I do not think this paper is ready for publication.

Detailed comments:
- Why is it sound to equate the expected risk over all clients with the expected risk over unparticipating clients? Doesn't that convolute the issues? It seems to me that there are two issues: the difference in risk between participating and unparticipating clients and the difference in risk between participating clients and all clients.
- Regarding generalization, there is work on quantifying the generalization gap in federated learning from heterogeneous datasets (cf. [2])
- Sec. 4.1.: How can you ensure that the pre-trained neural network captures semantically relevant features? How does the embedding vary with different networks?
- Sec. 4.1./App. D: Why does the dimension have to be further reduced, why is PCA the right method, why should it be reduced to exactly 256 dimensions, and why is it not better to train some neural network with a 256-dimensional representation right away?
- Sec. 4.1.: Why are mixtures created per label? How would the splits look if the mixtures where created independent of the label?

[1] Li, Xiaoxiao, et al. "FedBN: Federated Learning on Non-IID Features via Local Batch Normalization." International Conference on Learning Representations. 2021.
[2] Huang, Baihe, et al. "Fl-ntk: A neural tangent kernel-based framework for federated learning analysis." International Conference on Machine Learning. PMLR, 2021.

************* after rebuttal ******************

Given that I misunderstood a key concept in my original review and that we had an interesting discussion in this rebuttal period, I feel compelled to write a second review.

Summary Of The Paper:
In many federated learning applications, not all clients participate in the averaging process, but are, e.g., randomly sampled from the potentially infinite set of available clients, each endowed with its own local data distribution. This paper studies the decomposition of the generalization gap into the out-of-sample gap, i.e., the generalization ability of participating clients to unseen data drawn from their local distribution, and the participation gap, i.e., the difference between the generalization ability on unseen data drawn from the local distributions of participating clients to data drawn from all possible local distributions. The paper proposes a three-way-split of clients and local datasets to estimate these two quantities separately.

Main Review:
In many federated learning applications, local data distributions are not homogeneous. This fact has been studied extensively and methods have been proposed to address the difficulties in training on such heterogeneous distributions. In case the set of clients is large (or potentially infinite) and each client has a (potentially) different local data distribution, then at any given time, only a subset of those clients will have participated in federated learning. Current estimations of a federated learning system's generalization performance only assess it on data similar to the data seen by those participating clients. This overlooks the error on data drawn from a distribution from a client that not yet participated. While this error in general cannot be measured, it can be approximated if clients are drawn iid by holding out a subset of clients as "validation clients" to assess this participation gap. Naturally, this allows to also assess the level of heterogeneity in the local data distributions.

The main contribution of this paper hinges on the assumption of a potentially infinite number of clients, each endowed with its own, distinct local data distribution. Given this assumption, the proposed three-way-split is a straight-forward and sound (and fairly practical) method to assess the participation gap. This can indeed provide valuable insights about the performance and nature of the overall federated learning system.

Since standard benchmark datasets for non-iid distributions provide only a very limited number of different data distributions (e.g., the quintuple MNIST, MNIST-M, SVHN, USPS, and SynthDigits), the paper proposes a semantic partitioning of data. This allows one to simulate the setting of a very large set of clients each with a different local data distribution.

My main concern with the paper is that this contribution is not presented clearly enough and that its main assumption and limitations are not discussed in detail. Thanks to the great discussion with the authors, I am now convinced that the decomposition of the generalization gap into out-of-sample and participation gap has merits in the setting considered. It has also become clear that this is not so meaningful in case of a limited number of potential clients or a (small) finite number of potential distributions. Thus, in its current form the paper falls behind its potential.

I suggest the following improvements:
(i) Clearly point out the setting considered and its limitations. This includes separating it from cross-silo scenarios (where the number of potential clients is usually small) or many internet of things scenarios (where the number of potential types of devices is limited and devices of the same type share the same data distribution). I guess, the typical scenario is learning on mobile devices where the number of potential clients is indeed huge and local distributions are typically heterogeneous. Still, it would be great to discuss this assumption in a bit more detail: results from humanities (sociology and psychology) suggest that human behavior clusters around a fairly limited amount of archetypes, so it stands to reason that distributions obtained from human interaction with a device might also cluster around such archetypes. In the discussion with the authors, we established that in this case, the behavior or a federated learning system could quickly converge towards one with only a limited number of potential local data distributions.
While these discussions seem to limit the applicability of the paper, I think they greatly improve its significance, since it gives rise to interesting questions for future studies.
(ii) The empirical evaluation shows that the participation gap can be observed in practice. Here, the results on the naturally partitioned data are the most significant and should be pointed out more.
(iii) Since the results are not applicable to all federated learning settings, I suggest incorporating the community suggestions into a discussion section, such that these suggestions are properly contextualized within the assumptions and limitations of the approach.


These issues require some extensive changes to the manuscript that usually go beyond what is considered minor revisions. However, given that the discussion contained most of these points already, I am willing to trust that the authors can make it. The theoretical contributions remain straight-forward and the three-way-split is fairly natural. Thus, I am on the fence about this paper. Since the paper addresses a novel issue, I slightly tend towards acceptance. I have changed my score accordingly.


**Summary Of The Paper:**

In many federated learning applications, not all clients participate in the averaging process, but are, e.g., randomly sampled from the set of available clients. This paper proposes to study not only the generalization gap in federated learning, i.e., the difference between risk and empirical risk, but also the difference between the risk of clients that participate and those that do not participate in the current averaging round, which is called participation gap in this paper. The paper formally equates the risk of unparticipating clients with the expected risk over all clients, where the expectation is over the sampling probability of each client. The paper then proposes a set of data splits to estimate those gaps with a focus on heterogeneous local data distributions.

**Summary Of The Review:**

The issue of the different performance of participating and non-participating clients in federated learning - in particular for heterogeneous datasets - is interesting. The proposed contributions of this paper are a straight-forward data split strategy, a proposed strategy to split data into heterogeneous local datasets, and an empirical evaluation of the performance of participating and non-participating clients that confirms the intuitive expectations on the effect of participating clients. While the issue the paper addresses is interesting, the proposed contributions are fairly straight-forward and provide little novel insights. Thus, I vote for rejection.

---

> ### Author Response · Authors · 2021-11-13
> **Author Response to Reviewer bfMQ | Part 1 of 4**
>
> We thank the reviewer for their time and effort. We did notice there were major misunderstandings in the paper summary and throughout the main review, which we quote below.
>
> > ***The paper points out an often overlooked issue in federated learning: when clients are sampled for averaging, some clients may hold deprecated models. Those clients will have a worse performance than their recently averaged fellows. The paper shows that for homogeneous local data distributions, this error vanishes in expectation, but remains an issue for heterogeneous distributions. These insights are valuable.***
>
> This is **not** our claim at all. Our claim is not related to deprecated vs recently-updated models. The participation gap is defined by the performance gap on the **same, latest** model between the (unseen data of) participating clients and un-participating clients. This quantifies the generalization power to unseen clients, just as we care about generalization power to unseen samples in the traditional Empirical Risk Minimization framework. On the other hand, the gap between stale and latest model is much more trivial to justify, as they are from different convergence stages.
>
> Another major misunderstanding lies in the definition of participating clients (central to the setup of our paper).
>
> > ***...also the difference between the risk of clients that participate and those that do not participate in the current averaging round...***
>
> The participating clients are not defined as the clients of the current averaging round, but instead all the clients that have **ever** participated throughout the learning process. The unparticipating clients are defined by the clients that are never observed throughout the training process, which is also not related to the current round. This is why participation gap persists even with **centralized** training (see Figure 9), where sampling of clients across rounds is not involved.
>
> **Given these major misunderstandings of our work's setup and contributions, we would appreciate it if the reviewer could take another look at the paper.**  At this time it is not clear to us how to respond to the these concerns as they are critizing claims not made by our paper. We would be more than happy to respond to any follow-up questions.

---

> > ### Comment · Reviewer_bfMQ · 2021-11-15
> > **Response to Rebuttal**
> >
> > Dear authors,
> >
> > Thanks a lot for clarifying the contribution. I indeed misunderstood it. Thus, I will re-evaluate the paper entirely. To speed things up (and since we have this opportunity here), let me ask a few questions right away:
> > 1) Assume a finite number of distributions D_1,...,D_k from which all client distributions D_c are drawn from. Further assume that at some point, the participating clients cover all k distributions. Is it correct that the participation gap then remains constant, no matter how many more clients contribute?
> > 2) Given my now (hopefully) correct understanding of the participation gap, could you please clarify how this fundamentally differs from out-of-distribution generalization? On face value, the definitions seem very similar (even having multiple training distributions is standard in OOD generalization).
> > 3) How can intra-client splitting be used in practice? Intentionally holding out clients would degrade the overall performance. Not using clients for communication reduction would make testing on them infeasible.
> >
> > Cheers

---

> > > ### Author Response · Authors · 2021-11-16
> > > **Author Response to Follow-up Questions of Reviewer bfMQ | Part 1 of 3**
> > >
> > > Thank you for your prompt response! We are happy to learn that you are willing to re-evaluate our paper. We hope you enjoyed reading it. We also appreciate your follow-up questions --- these are all great questions, and we will incorporate our responses in our next revision.
> > >
> > > > ***Q: Assume a finite number of distributions D_1,...,D_k from which all client distributions D_c are drawn from. Further assume that at some point, the participating clients cover all k distributions. Is it correct that the participation gap then remains constant, no matter how many more clients contribute?***
> > >
> > > - Before getting into the question, we first identify that restricting the possible universe of local distributions to a **finite** set may **not** be the ideal or most realistic model (at least not in cross-device FL). This finite distribution model effectively restricts the "types" of clients to $k$: by pigeonhole principle, any $k+1$ clients will have at least two clients sharing **exactly the same distribution**. This is unlikely to be true in FL practice (e.g., modeling user preferences). **This model is only true if one is certain that the "types" of clients are limited to $k$, no matter how many clients are collected in the future.**
> > >
> > > Back to your question. The short answer is **no**, the participation gap will converge to 0 as the number of participating clients increases to infinity, instead of remaining constant. We explain as follows (the explanation may be counterintuitive due to the finite-$k$ restriction, but it would be easier to understand if one thinks of $k$ as sufficiently large.)
> > >
> > >
> > > Suppose we assume the "types" of clients are limited to $k$ and the meta-distribution across these $k$ distributions is **multinomial**. At the time the participating clients cover all $k$ distributions, at the client level, one only has an empirical approximation of the meta-multinomial-distribution. The participation gap **still exists** due to the mismatch of the empirical approximation of multinomial meta-distribution, and the real multinomial meta-distribution.
> > >
> > > (To see this, consider approximating a multinomial distribution ($p_1$, $p_2$, ... $p_k$). Suppose you observe some ($n_1$, $n_2$, ..., $n_k$) by i.i.d. drawing from the multinomial distribution, and at some time all $n_i \geq 1$. At this time, the estimated $\hat{p}_i = \frac{n_i}{\sum n_i}$ can still be very different from the real $p_i$. The $\hat{p}_i$ will converge to $p_i$ if and only if the sampling size $N = \sum n_i \to \infty$. The larger $k$ is, the slower the convergence is.)
> > >
> > > As the number of participating clients increase, the empirical approximation of the meta-distribution will converge to the real multinomial meta-distribution, and the participation gap will converge to 0. Again, one should interpret this result with care as the finite $k$ assumption may be unrealistic

---

> > > ### Author Response · Authors · 2021-11-16
> > > **Author Response to Follow-up Questions of Reviewer bfMQ | Part 2 of 3**
> > >
> > > > ***Q: Given my now (hopefully) correct understanding of the participation gap, could you please clarify how this fundamentally differs from out-of-distribution generalization? On face value, the definitions seem very similar (even having multiple training distributions is standard in OOD generalization).***
> > >
> > > We briefly discussed the relation in §1.1 of our manuscript. The short answer is that our settings are **related** to but **different** from OOD robustness. The key difference is that in our setting, the distribution of an unparticipating client is **not completely** out-of-distribution. With in each client, the local distributions are different; across clients, the meta-distribution is shared. In an OOD setting, there is not such an outer consistency. We discuss more formally as follows.
> > >
> > > We start by discussing the similarity of these two settings (FL and OOD). In both cases, we attempt to train on some distribution of data $p(x)$ and then considers the implications of deploying on a different distribution $\grave{p}(x)$.  Our problem is the specific case of this where $(c, \alpha_c), (\grave{c}, \alpha_{\grave{c}}) \sim p(c, \alpha_{c})$, $p(x) = \sum_{c} \alpha_c p(x | c)$, and $\grave{p}(x) = \sum_{\grave{c}} \alpha_{\grave{c}} p(x | \grave{c})$, that is to say our set of clients and their respective weight of evidence are generated IID from the client distribution $p(c, \alpha_c)$, but where the clients have different data generating distributions (i.e. conditioned on $c$). Because the clients are drawn IID from meta-distribution, as the number of clients becomes very large, the participating and unparticipating clients converge to the same distribution.
> > >
> > > This is different from the typical OOD generalization setup because those cases assume $c$ and $\grave{c}$ are not drawn from the same distribution (i.e. Federated MNIST vs. Federated SVHN), and they further assume that many (or most) of the test examples are drawn from a different distribution than the training distribution (i.e. $x\sim\grave{p}(x)$ are outside of the typical set of $p(x)$).  Both assumptions mean that as the number of participating (non-OOD) and unparticipating (OOD) clients becomes very large, they do _not_ converge to the same distribution.
> > >
> > > Consequently, while we show that increased participation in training leads to a reduction in the participation gap (and therefore presents an incentive for increasing the amount of participation in federated training), it does not lead to better OOD generalization.  The efforts to do OOD generalization (either by explicit OOD detection or by applying more robust calibration to the models) are largely independent to the efforts to improve generalization to new clients in cross-device FL.
> > >
> > > Finally, note that there is also future work that could combine elements from both topics.  In particular, OOD detection methods could be used to identify the specific examples that contribute most significantly to the participation gap, from which we could either use those examples in future rounds of training to reduce the gap, or just refrain from making predictions on those examples in order to reduce the gap for the data on which we actually serve predictions.  Unfortunately, while it would reduce the participation gap, the second of these is likely sub-optimal since it could enhance bias and reduce fairness by refusing to make predictions on clients whose data is underrepresented in the full population.

---

> > > ### Author Response · Authors · 2021-11-16
> > > **Author Response to Follow-up Questions of Reviewer bfMQ | Part 3 of 3**
> > >
> > > > ***Q: How can intra-client splitting be used in practice? Intentionally holding out clients would degrade the overall performance.***
> > >
> > > (**Intra-client** splitting refers to splitting the dataset within a client, but the remaining question seems to ask about **inter-client** splitting (hold out certain unparticipating clients). We assume the question is on **inter-client** splitting, but feel free to post any follow-up questions.)
> > >
> > > Also, to avoid misinterpretation, we consider two possible interpretations of the term "in practice": 1) in simulation environment (benchmarking/prototyping/research), and 2) in production environment (when one trains and serves on real clients.)
> > >
> > > 1. In the simulation environment (e.g., when one evaluates, benchmarks, or prototypes federated algorithms), we would recommmend always applying the three-way split by default. As discussed in §3.2, evaluating participation gap enables researchers to quantify client heterogeneity, measure overfitting to population distribution, quantify (and predict) robustness to unseen clients, and quantify the incentive for clients to participate. We agree that this reduces the amount of data used for training, but is necessary for benchmarking real-world performance (where we care about algorithms' performance on unseen clients).
> > >
> > > (As a side note, again, this is akin to the classic ML setting. Does holding out validation data (instead of training on all data) reduce the performance? Yes. But the held-out validation data gives additional insights on the model performance beyond training data.)
> > >
> > > The exact held-out fraction of unparticipating clients should be determined by balancing the overall performance with the variance of unparticipating metrics. For large scale experiments, we observed a 5\% held-out rate would be sufficient to provide reasonable un-participating metrics. We also stress that one does not have to hold out un-participating clients every communication round.
> > >
> > > 2. In the production environment, the simplest approach to perform inter-client splitting is to select from only a subset of clients (participating clients) for training and select from a separate subset of clients (un-participating clients) for testing. In a large-scale setting, this may actually have no impact on performance, since the number of clients sampled in a round (e.g. hundreds) for either training or testing may be much smaller than the population size (e.g. millions), and so most clients will not participate in training anyway.
> > >
> > > ---
> > >
> > > > ***Q: Not using clients for communication reduction would make testing on them infeasible.***
> > >
> > > Please correct us if we are misunderstanding your question, but holding out clients as part of the three-way split is not intended to reduce communication. In a large-scale setting, we can sample the same number of users from a population for training each round, regardless of whether we perform the inter-client split. In the simplest setting, the server could just send the model to the unparticipating clients, evaluate on their local data, and return the metrics (as unparticipating performance). Hopefully this helps, let us know if you have further questions.
> > >
> > > ---
> > >
> > > Thank you again for your willingness to re-evaluate the work – your suggestions have significantly improved the clarity of the paper.

---

> ### Author Response · Authors · 2021-11-13
> **Author Response to Reviewer bfMQ | Part 2 of 4**
>
> We address other points in detail:
>
> > ***Q: The semantic partitioning of data into heterogeneous local datasets is interesting, but the advantage of the method over other strategies (simple distance based clustering, or using semantically similar datasets (e.g., MNIST, MNIST-M, SVHN, USPS, and SynthDigits)) remains unclear.***
>
> This is a great question! The goal of semantic synthetic partitioning is to partition a centralized dataset (e.g., CIFAR-10, MNIST, ImageNet) without client identity into semantically clustered clients **while satisfying the FL problem formulation** in §3 (data is modeled as a distribution of distributions). Using similar datasets for different clients as suggested by the reviewer does not have this property (we go deeper into this below), which is well-motivated by real-world data.
>
> Our approach enables us to create a more "natural" synthetic dataset from standard benchmarks, and makes it easier to perform federated research on standard baselines. This also enables the federated research community to study the effect of heterogeneity (by, for example, synthesizing federated datasets with various levels of heterogeneity). See our newly introduced semantic CIFAR-100 and MNIST datasets described in Appendix D and our description of our open-source code in §7. As mentioned in our paper, there are numerous existing works on creating federated datasets from standard centralized datasets, but to the best of our knowledge most of the existing works are devoted to creating label heterogeneity.
>
> ---
>
> > ***Q: Why not simple distance based clustering?***
>
> Simple distance based clustering is too sensitive to semantically irrelevant features (such as translation). In an extreme case, an image translated by 1 pixel can have a very large $\ell_2$ distance from the original one. We would intuitively want clusters to be more focused on useful features (such as writing styles, etc.). This is the motivation of using a pretrained embedding.
>
> ---
>
> > ***Q: Why not using semantically similar datasets (e.g., MNIST, MNIST-M, SVHN, USPS, and SynthDigits)?***
>
> Combining various semantically similar datasets into a federated dataset can be useful (e.g., cross-silo), but is limited for the following reasons.
> 1) Unless we can further partition each dataset (as we proposed in our paper), the number of possible heterogeneous distributions (i.e., clients) is strictly limited by the number of datasets available. The latter is usually low, as it is hard to find a large number of datasets (not single data point) that are similar to MNIST, for instance. It is not clear how to simulate, say, $10^3$ heterogeneous distributions just by merging datasets. In summary, we need an approach to semantically **divide** the dataset into subpopulations, in addition to just **addition**.
> 2) Developing a semantic division of standard datasets (e.g., CIFAR, ImageNet) also facilitates comparison of FL research with centralized research.
> 3) Merging multiple datasets complicates the setup as it may not satisfy the same meta-distribution-assumption imposed in §3. This is more like a testbed for domain adaptaion (or OOD generalization), instead of out-of-client generalization. Even if one wants to study OOD generalization, we believe it is still necessary to disentangle the in-domain generalization first, as proposed in this work.
>
> ---
>
> > ***Q: ...Using naturally partitioned datasets is fairly common in personalized FL or FL from heterogeneous data sources (e.g., in FedBN [1])... The semantic partitioning might be an interesting new technique, but from this paper its benefits remain unclear.***
>
> We thank the reviewer for suggesting this work! We have incorporated it into our related work section (see §A).
>
> We agree that using natural datasets is common, but in the literature it is also very common (and sometimes necessary) to synthesize federated datasets. We listed various motivations above and in the paper (e.g., enable comparing with standard centralized benchmarks, enabling ablation study on heterogeneity level, etc.). We suggest that in these cases, it is important to incorporate other more natural forms of heterogeneity in addition to *just* label heterogeneity, since as we show in Figures 4 and 5, label-based heterogeneity is not enough to produce realistic generalization behavior.

---

> ### Author Response · Authors · 2021-11-13
> **Author Response to Reviewer bfMQ | Part 3 of 4**
>
> > ***Q: Suggesting to report the distribution of metrics seems particularly odd, since this is common practice in most ML papers (at least, reporting standard deviation or variance).  In FL, the variance of local client performances is also often reported, or visually illustrated.***
>
> We are suggesting to report the distribution of metrics **across clients**, not across runs. Our main motivation of this suggestion is that the distribution of clients **across metrics** can be **very different for participating validation and unparticipating clients**. For instance, for the EMNIST-10 experiment in table 2, we showed that the unparticipating clients have much heavier tail distribution (91.7\% vs 100.0\% accuracy for the lowest 5th percentile). We encourage researchers to pay attention to the difference of two distributions (participating and unparticipating) as it may have fairness implications. We updated our manuscript to clarify this suggestion (see §6).
>
> ---
>
> > ***Q: Only the first suggestion - to analyze the participation gap - seems appropriate, but is limited to approaches where clients are sampled and - as the experiments suggest - the sampling probability for individual clients is sufficiently low.***
>
> This claim is not correct and is caused by the misunderstanding of the definition of participation gap, as mentioned in the beginning of the response. The participation gap is **not limited to approaches where clients are sampled**, and has nothing to do with sampling during rounds. Even for centralized optimizers or federated optimizers with no sampling the participation gap still exists.
>
> ---
>
> > ***Q: Why is it sound to equate the expected risk over all clients with the expected risk over unparticipating clients? Doesn't that convolute the issues?***
>
> The answer to this question follows from the definitions of participating and unparticipating clients (§3), fundamental to the setup of our work. Under the proposed definitions, expected risk is the only way to characterize unparticipating clients, similar to how we use validation data in classic ML to understand expected risk on unseen data. We provide more detail below.
>
> In our definition, unparticipating clients refer to the (infinite) set of clients that never participate in training. This is akin to the validation set in plain train/val split --- the validation accuracy corresponds to the accuracy on a fresh sample drawn from the distribution, and thus expected risk. In FL setting, each client has a local sub-distribution. The unparticipating accuracy is the accuracy of drawing a random client from meta-distribution, and then a sample from its local distribution, and thus expected risk.
>
> ---
>
> > ***Q: It seems to me that there are two issues: the difference in risk between participating and unparticipating clients and the difference in risk between participating clients and all clients.***
>
> This is not quite right. We define unparticipating clients to be drawn from the possibly infinite set of clients. In other words, unparticipating clients are just all clients in the distribution sense. On the other hand, the participating clients are a **finite** set of clients that actually participate in training.
>
> This scenario is akin to the classic ML: if one wants to predict the performance on unseen data, one has to hold out some data as validation/test data. In FL, in order to predict the performance on un-participating clients, one has to hold out some clients as "validation" clients to evaluate. We name it "un-participating" mainly to avoid overloading the terminology "validation", as there is another level of train/val split (participating training and participating validation).
>
> ---
>
> > ***Q: Regarding generalization, there is work on quantifying the generalization gap in federated learning from heterogeneous datasets (cf. [2])***
>
> We thank the reviewer for suggesting this reference. We have incorporated it into our related work section (see §A)). However, [2] does not disentangle the out-of-sample gap and generalization gap, which is central to our contributions.

---

> ### Author Response · Authors · 2021-11-13
> **Author Response to Reviewer bfMQ | Part 4 of 4**
>
> > ***Q: Sec. 4.1.: How can you ensure that the pre-trained neural network captures semantically relevant features? How does the embedding vary with different networks?***
>
> This is a good question. We cannot ensure the pre-trained neural network captures semantic features, but it is commonly observed by the deep learning community (e.g., computer vision, transfer learning) that pretrained embeddings (often taken from penultimate layers of classifiers) capture useful semantic features for downstream tasks. Regarding how the embedding varies with different networks: we observe that pretrained EfficientNetB3 generally performs better in producing intuitive clusters than smaller architectures such as ResNet-18 and EfficientNetB0.
>
> ---
>
> > ***Q: Sec. 4.1./App. D: Why does the dimension have to be further reduced, why is PCA the right method, why should it be reduced to exactly 256 dimensions, and why is it not better to train some neural network with a 256-dimensional representation right away?***
>
> Reducing the dimension is purely a computational issue since the original embedding (>1k) is too large for downstream procedures such as GMM fitting and optimal matching (measured by KL divergence). We added more descriptions in §D. Thank you for the great question!
>
> ---
>
> > ***Q: why is PCA the right method, why should it be reduced to exactly 256 dimensions***
>
> PCA intends to capture top dimensions of variation of the semantic pretrained embeddings, which is intuitive in this context. PCA may not be the best method, but we found PCA to be simple enough to generate reasonable results. The dimension of 256 is a trade-off of (down-stream) computational complexity and embedding information.
>
> ---
>
> > ***Q: Why is it not better to train some neural network with a 256-dimensional representation right away?***
>
> This could be better, but most off-the-shelf (moderate to large-scaled) pretrained networks are released with more than 1000 dimensions. We wish to leverage an off-the-shelf, pre-trained embedding so that we don't have to train the embedding again. This significantly reduces carbon footprint and enables using this method for a wider range of federated researchers.
>
> ---
>
> > ***Q: Sec. 4.1.: Why are mixtures created per label? How would the splits look if the mixtures where created independent of the label?***
>
> This is a good question. We added a more detailed discussion in the latest version of our paper (see §4). A summary:
>
> If one simply generates the mixture for all labels, then the similar labels will be grouped together, meaning that the clusters are largely correlated to labels. This is especially the case when the pre-trained embeddings are sufficiently powerful. Since our goal is to analyze semantic heterogeneity, we have to disentangle the effect of semantic heterogeneity from label heterogeneity. To this end, we instead impose intra-client intra-label similarity and intra-client inter-label similarity in two separate stages.
>
> We wish to highlight that one could still synthesize label heterogeneity on top of our semantic partitioning framework (if one wants to). Our main contribution is to provide a clean way to distill and analyze the semantic heterogeneity.
>
> ---
>
> We thank the reviewer again for their comments. We ask that in light of these detailed clarifications, the reviewer might re-evaluate our paper and consider adjusting their initial rating.

---

### Official Review · Reviewer_TBV6 · 2021-11-06

**Correctness:** 4
**Technical Novelty And Significance:** 3
**Empirical Novelty And Significance:** 2
**Recommendation:** 6
**Confidence:** 3

**Main Review:**


Strengths

-The paper analyzes performance measurements in the popular federated learning paradigm. The proposal is sensible and addresses an important problem.
-The experiments are thorough and use best practice
-The authors discuss releasing a code base for running their experiments

Weakness/Comments

-The experiments are done using FedAVGM. First of all this is not even mentionedi n the main paper. I think it would be interesting for the proposed study to revisits how different methods perform under the evaluation protocol and also use SOTA methods (e.g. FedYogi), perhaps conclusions might be different about which method is best.
-In some federated settings the distribution shift is in the labels, how do the authors consider this case in ther analysis of non participating clients. For example a natural non-participating client might have a label unseen in other clients, yet the model is relevant,
-(Minor) Figure 2 makes it hard to notice the per column similarity in the digits (and thus the distribution shift). I would suggest to instead do something more obvious (different color digit or using a different digit in each row altogether)


**Summary Of The Paper:**

The authors propose a different view of performance in federated learning, by measuring separately the performance out of sample  and the performance on novel (but related distributions) from a “non-participating” client. They propose one approach of structuring experiments that allows these kins of measurements using a three way split. They then study how the dataset, client number and diversity affects the perforamnce.


**Summary Of The Review:**


Overall the paper addresses important topic and has conclusions that could benefit the community. The experiments seem too be well done and rigorous.

---

> ### Author Response · Authors · 2021-11-13
> **Author Response to Reviewer TBV6**
>
> We thank the reviewer for their time and for their valuable suggestions. We're glad to see the reviewer appreciated the importance of the problem we tackle, the thoroughness of our experiments, and our open-source code framework enabling researchers to easily implement our community suggestions. We address all concerns below. We're happy to answer any follow-up questions.
>
> > ***Q: -The experiments are done using FedAVGM. First of all this is not even mentionedi n the main paper. I think it would be interesting for the proposed study to revisits how different methods perform under the evaluation protocol and also use SOTA methods (e.g. FedYogi), perhaps conclusions might be different about which method is best.***
>
> We actually applied (and reported) **FedAdam** (also proposed in Reddi et al. 2021) for two NLP tasks (Shakespeare, Stackoverflow). We also experimented with **FedYogi** on these tasks, and the performance is comparable (in terms of both participating validation and unparticipating metrics). We applied FedAvgM on two vision tasks (EMNIST, CIFAR) since it performed better than FedAdam/FedYogi. We also experimented with vanilla FedAvg and FedAdagrad, which are less effective than the other adaptive optimizers, but the participation gaps are generally consistent. We also extensively experiment with different hyperparameter choices, varying the computation/communication tradeoff, adding weight decay, and varying model architecture in Appendix C. We generally observe a consistent participation gap across these variations.
>
> As a side note, we would like to point out that the participation gap still persists even with centralized optimizers (see Figure 9). This is different from the convergence efficiency issue, also caused by heterogeneity, which can be fully resolved by training with centralized optimizers. Most of the FL literature tackling heterogeneity so far are devoted to this convergence issue, e.g., Adaptive Federated optimizers, FedProx, SCAFFOLD, and FedDyn. In the extreme case, as long as these federated optimizers converge to the same point (which is the case for convex models and sufficiently small learning rate), they should experience the same participation gap. As mentioned above, we indeed observe consistent participation gaps across various tested optimizers (FedAvg, FedAvgM, FedAdam, FedYogi, etc), which suggests a certain level of optimizer agnosticity of participation gap.
>
> That said, we will definitely add the optimizer details in the main paper (and thanks for reminding us!).
>
> ---
>
> > ***Q: In some federated settings the distribution shift is in the labels, how do the authors consider this case in ther analysis of non participating clients. For example a natural non-participating client might have a label unseen in other clients, yet the model is relevant.***
>
> Great question! The setup of our paper assumes that the un-participating clients and participating clients are both drawn from the **same** meta-distribution, though each client may have different local distribution of data. This setup could still incorporate label inconsistency.
>
> If one instead drops the same meta-distribution assumption, and assumes the participating clients and un-participating clients are from two different meta-distributions, one can still formally apply the three-way splits to out-of-distribution clients (with a different meta-distribution). However, it seems the better option is to apply a four-way split (original three-way split + an OOD clients) to disentangle the effect of participation gap and the OOD effect. Note that this modified setup is likely to be less tractable from both an empirical and theoretical perspective, since no samples from the second meta-distribution are seen during training.
>
> ---
>
> > ***Q: (Minor) Figure 2 makes it hard to notice the per column similarity in the digits (and thus the distribution shift). I would suggest to instead do something more obvious (different color digit or using a different digit in each row altogether)***
>
> Please correct us if we misunderstood the question, but the coloring in Figure 2 illustrates the three-way split scheme. The blue, orange, purple colors correspond to *participating training*, *participating validation*, and *unparticipating* datasets. We only present the digit "6" to make the similarity more apparent (note that within each column (client), the styles of "6" are very similar). Would the reviewer mind elaborating what makes it hard to notice the per column similarity in Figure 2? We're happy to make any changes. Thanks in advance!
>
> ---
>
> We thank the reviewer again for their comments, they have helped us improve our work.

---

### Author Response · Authors · 2021-11-30
**Thank you for your comments!**

We thank all of the reviewers for their valuable suggestions. We've addressed each reviewer's comments below in detail and appreciated the lively discussion, which has significantly strengthened our work. We have incorporated the suggestions and our response in the latest revision.

We're happy to answer any further questions from the reviewers or AC. We noticed Reviewer dKHn has not yet participated in the rebuttal process; we hope that our detailed comment clarified any misunderstandings.

We'd again like to thank all the reviewers and the AC for their time and effort.

---

### Decision · Program_Chairs · 2022-01-20

**Decision:**

Accept (Poster)

**Comment:**

This paper presents some insightful suggestions for researchers studying generalization in federated learning by separating two types of performance gaps between training and test performance, the participation gap (due to partial client participation) and the performance gap (due to data heterogeneity). They suggest that federated learning researchers use a three-way split between participating clients' training data, participants clients' validation data, and non-participating clients' data to measure the generalization performance of an FL model. The paper presents thorough experiments to support their conclusions. A common concern about the paper is that the authors' suggestions, although relevant and reasonable, are somewhat unsurprising and have been noted in different forms in other works in federated learning. Another concern is that the conclusions are purely based on experiments and are not supported by theoretical justification. Despite these concerns, the reviewers commended the overall insights presented in the paper.

There was a healthy post-rebuttal discussion and some reviewers reevaluated the paper and raised their initial scores. Therefore, I recommend acceptance of the paper. I encourage the authors to take the reviewer's constructive suggestions into account when preparing the final version of the paper.